# Lipid-Based Nanocarriers for Neurological Disorders: A Review of the State-of-the-Art and Therapeutic Success to Date

**DOI:** 10.3390/pharmaceutics14040836

**Published:** 2022-04-11

**Authors:** Bwalya Angel Witika, Madan Sai Poka, Patrick Hulisani Demana, Scott Kaba Matafwali, Siyabonga Melamane, Sandile Maswazi Malungelo Khamanga, Pedzisai Anotida Makoni

**Affiliations:** 1Department of Pharmaceutical Sciences, School of Pharmacy, Sefako Makgatho Health Sciences University, Pretoria 0208, South Africa; madan.poka@smu.ac.za (M.S.P.); patrick.demana@smu.ac.za (P.H.D.); 2Clinical Research Department, Faculty of Infectious and Tropical Diseases, London School of Hygiene & Tropical Medicine, London WC1E 7HT, UK; esco1789@gmail.com; 3Stutterheim Hospital, No.1 Hospital Street, Stutterheim 4930, South Africa; cyamelamane@gmail.com; 4Division of Pharmaceutics, Faculty of Pharmacy, Rhodes University, Makhanda 6140, South Africa; s.khamanga@ru.ac.za; 5Division of Pharmacology, Faculty of Pharmacy, Rhodes University, Makhanda 6140, South Africa

**Keywords:** central nervous system, blood-brain barrier, brain-targeted drug delivery, lipid-based nanocarriers, neurodegenerative diseases

## Abstract

Neurodegenerative disorders including Alzheimer’s, Parkinson’s, and dementia are chronic and advanced diseases that are associated with loss of neurons and other related pathologies. Furthermore, these disorders involve structural and functional defections of the blood-brain barrier (BBB). Consequently, advances in medicines and therapeutics have led to a better appreciation of various pathways associated with the development of neurodegenerative disorders, thus focusing on drug discovery and research for targeted drug therapy to the central nervous system (CNS). Although the BBB functions as a shield to prevent toxins in the blood from reaching the brain, drug delivery to the CNS is hindered by its presence. Owing to this, various formulation approaches, including the use of lipid-based nanocarriers, have been proposed to address shortcomings related to BBB permeation in CNS-targeted therapy, thus showing the potential of these carriers for translation into clinical use. Nevertheless, to date, none of these nanocarriers has been granted market authorization following the successful completion of all stages of clinical trials. While the aforementioned benefits of using lipid-based carriers underscores the need to fast-track their translational development into clinical practice, technological advances need to be initiated to achieve appropriate capacity for scale-up and the production of affordable dosage forms.

## 1. Introduction

Over the years, advances in medicines and therapeutics have led to an enhanced understanding of various modes of action in targeted drug therapy to the central nervous system (CNS). This has been mostly attributed to the modification of signaling mechanisms, such as the activation of the PI3K/Akt pathway after an ischemic stroke attack and interruption of α-synuclein aggregation to avert neuronal degeneration in patients with Parkinson’s disease [1,2].

Nevertheless, therapeutic options for delivery to the CNS are marred by factors such as the presence of the blood-brain barrier (BBB), inadequate drug concentrations reaching neuronal tissues and/or inadequate residence time of drug(s) with the corresponding receptor(s). As such, various approaches have been proposed or used to circumvent these shortcomings at both the macroscopic and microscopic level [3].

Owing to this, lipid-based nanocarriers, broadly categorized as bilayered vesicular drug delivery systems, unilayered lipid drug delivery systems, and submicron emulsions, have been developed in an attempt to overcome shortcomings related to CNS delivery of therapeutic molecules and are discussed herein [2,4]. Over the years, various researchers have looked into the application of lipid-based nanocarriers for the treatment of different neurological conditions. In particular, more recently (over the past 2–3 years), Akel et al. reviewed on the progress and perspectives of brain-targeted lipid-based nanosystems via the nasal route in Alzheimer’s disease [5], Nsairat et al. reviewed lipid nanostructures for targeting brain cancer [6], and Ali et al. reported on nanotechnological advances in the treatment of epilepsy [7]. Furthermore, more broadly, Shilpi et al. reported on lipid nanocarrier-based drug delivery for the treatment of brain-related disorders via various delivery routes [8]. Of closer significance to the literature review presented herein is the report by Nguyen et al. on lipid-based nanocarriers via the nose-to-brain pathway for CNS disorders. In their paper, the authors focused on advances in research on liposomes, solid lipid nanoparticles, nanostructured lipid carriers, microemulsions, nanoemulsions, and cubosomes in managing some neurological disorders [9].

Distinguishably, in this review, focus is placed on the state-of-the-art of current applications of nano lipid drug delivery systems, including liposomes, pharmacosomes, ethosomes, transferosomes, niosomes, solid lipid nanoparticles, nanostructured lipid carriers, lipid drug conjugates, nanocapsules, micelles, self-nanoemulsifying drug delivery systems, microemulsions, and nanoemulsions, in therapy and diagnosis of the most common diseases affecting the CNS, as well as highlighting future nanotechnological approaches and perspectives. 

## 2. Anatomy and Physiology of the CNS

### 2.1. Barriers to CNS Drug Delivery

The CNS architecture itself limits the ability of most drugs to reach the CNS compartments in significant amounts [10]. A drug must cross several barriers to achieving a desired therapeutic level in CNS. Many drugs are discarded at clinical trials due to their limited entry to the brain parenchyma [10,11]. There are several barriers to effective drug delivery such as the blood-brain barrier (BBB), the blood-cerebrospinal fluid barrier (BCSFB), and the blood-brain tumor barrier, and efflux proteins [12] can also impede the delivery of drugs [10,13,14]. The BBB represents the main challenge for drug delivery into the CNS microenvironment. The presence of tight junctions and other specialized cells in the barrier impedes the paracellular flux of intended therapeutic molecules by posing high electrical resistance [15].

#### 2.1.1. The Blood-Brain Barrier (BBB)

The BBB plays a role in the regulation and proper functioning of the brain, in so doing keeping peripheral circulation separate from CNS. The keeping of peripheral circulation separate from the CNS offers protection from the entry of dangerous substances into the brain. The BBB performs filtering functions and is selectively permeable to substances such as water, nutrients, and hydrophobic molecules through passive transport [16]. The BBB has a protective mechanism that involves P-glycoprotein (P-gp) mediated efflux of lipid-soluble toxins and bacteria [17]. In addition, the BBB also keeps the fluid level of the brain at the required volume and prevents the frequent transport of fluid and salt to the extracellular matrix [16,17]. The BBB comprises cells such as the astrocytes and endothelial cells that are joined together to form tight junctions (JTs) [18]. The astrocytes in the brain are also critical for the proper functioning of the BBB and provide support to the endothelial cells of the BBB [19]. The tight junctions are basic structural components essential for the functional activity of endothelial cells. Normal capillaries of the blood circulation are deprived of these tight junctions of endothelial cells, but the tight junctions of BBB endothelial cells protect the brain [18]. These tight junctions consist of transmembrane proteins such as claudins, occludins, junctional adhesion molecules (JAM), and endothelial cell adhesion molecules (ESAMs) [18,20]. Endothelial cells bind to subunits of tight junction via a complex of protein known as zo-1 and associated proteins [21]. 

#### 2.1.2. The Blood-Cerebrospinal Fluid Barrier (BCSFB)

The blood-cerebrospinal fluid barrier (BCSFB) contributes to CNS complexity and adds an additional barrier for therapeutics. A group of ependymal cells forms the choroid plexus, and this choroid plexus operates as a barrier in the brain by separating the blood from cerebrospinal fluid (CSF) [22]. The ependymal cells on the surface of the brain are folded over onto themselves to form a double layer which is called the arachnoid membrane [22]. The arachnoid membrane is selectively permeable to hydrophilic substances. The choroid plexus and the arachnoid membrane act jointly at the barrier between the blood and the CSF [14,22]. The BCSFB is also strengthened by an active organic acid transport system in the choroid plexus, which is capable of driving CSF-borne organic molecules into the blood [23]. As a result of this mechanism, several drugs are actively removed from the CSF and in turn inhibited from crossing into the brain parenchyma [14]. The drugs exhibit poor lipid solubility and are therefore restricted by the tight junctions (TJs) between the cells; this forms a barrier that requires additional assistance for their transport across the BBB.

#### 2.1.3. The Blood-Brain–Tumor Barrier (BBTB)

The normal functions of the BBB are altered during different disease states of the brain [24]. In some disease states, such as COVID-19, stroke, dementia, brain tumors, and multiple sclerosis, the BBB loses its functions to some extent and permits the entry of microorganisms and toxins to the brain [24,25,26,27]. Brain tumors such as gliomas that primarily originate from glial tissues are mainly characterized by the loose tight junctions of endothelial cells [28,29]. Low-grade gliomas keep the normal functioning of the BBB, whereas high-grade gliomas have altered permeability of brain vasculature [30]. High-grade gliomas are characterized by enhanced angiogenesis and vascular endothelial growth factor (VEGF) that fulfils the demand for accelerated metabolism. The enhanced angiogenesis and VEGF lead to the formation of blood vessels and BBTB with altered functions [28,30]. Several reasons may be responsible for the altered permeability of endothelial cells. The astrocytes in the glioma brain are not well differentiated and unable to secrete the required chemicals for the proper functioning of the BBTB; this then leads to leaky blood vessels [30]. The expression of claudins has the potential to contribute to leaky endothelial junctions as well. Some studies have reported that downregulation or loss of claudins and occludins enhanced the permeability of the BBTB [31,32]. The downregulation of claudin-3 and claudin-5 expressions and loss of claudin-1 led to leakage of tight junctions [31]. The loss of expression of these proteins in astrocytes and metastasis of adenocarcinoma has also been shown in studies [30,33]. The studies have shown that astrocytoma and other brain tumors secrete VEGF, cytokines, and hepatocyte growth factors that downregulate the expression of occludins [34,35]. There is still more research needed to explore the exact mechanism through which claudins damage the BBTB. Therefore, the BBTB is a barrier to providing the effective therapy of brain tumors and leading to a subtherapeutic level of chemotherapeutic agents in brain tumor tissues.

#### 2.1.4. Efflux Transporters

The presence of the BBB and different efflux transporters contribute significantly to the poor penetration of drugs to the brain and contributes to the resistance of tumors. Various efflux transporters that depend on ATP are present in the BBB. These efflux transporters are mostly found on the luminal membranes of the epithelial cells in the BBB and function as clearance systems for metabolites and catabolites produced in the brain [12]. The efflux transporters in the BBB are transmembrane P-glycoprotein (P-gp) and multidrug resistance-associated protein (MRP) that are both classified as ATP-binding cassette (ABC) transporters because they use ATP hydrolysis to translocate substrates across their concentration gradient [17]. P-gp is encoded by the multidrug resistance gene MDR1 and is the main drug efflux mechanism of the BBB. The P-gp transporter transports many diverse classes of substances including drugs out of the brain [36]. Additionally, some cancer cells have been shown to express large amounts of P-gp, MDR-associated protein 1 (MRP1), and breast cancer resistance protein (BCRP), further amplifying that efflux effect and rendering these cancers multidrug-resistant (MDR) [37,38]. Some drugs and other substances that escape the P-gp clearance mechanism are later pumped out of the brain parenchyma via a second mechanism, which is termed phase I and II metabolisms. In phase I modification, enzymes such as alkaline phosphatase and cytochrome p450 either uncover or add reactive polar groups to the surface of the drug molecule; this is then followed by the conjugation of anionic groups onto the drug molecules as part of phase II modification with an enzyme such as glutathione S-transferase (GST) in readiness for expulsion [39,40,41]. The phase I and II metabolism reduce the lipid solubility of the drug molecule and in turn inhibits the passage of compounds back into the brain parenchyma.

In addition to P-gp, MDR proteins (MRPs) which belong to the ABC family are responsible for the transport of anionic substances, whereas P-gps are mainly responsible for the efflux of cationic substances from the brain [12]. Some drugs, however, are transported by both proteins. Some of the MRPs are present in the luminal membrane of the endothelial cells which is an ideal place for the proper functioning of the efflux transporter. Breast cancer-related proteins (BCRP) are found to be present in the apical membrane of the BBB endothelial cells and expectedly responsible for the inhibition of brain uptake of some potential drugs [37,42]. Transporters of the organic anion transporting polypeptide family (OATP-family) and the organic anion transporter-family (OAT-family) are also located in the brain and may be involved in the efflux of drugs [43]. These transporters work alongside the concentration gradient. OATP and OAT transporters are ATP independent reversible transporters and depend on the concentration of drug [43,44].

### 2.2. Nose-to-Brain Drug Delivery

Prior to direct-to-brain delivery, the active pharmaceutical ingredients (API) must be transported across the nasal epithelium to reach the lamina propia consisting of two cranial nerves, trigeminal, and olfactory nerves, which are primary connections between the nasal cavity and the CNS [45]. The API is subsequently directly transported to the brain along olfactory and trigeminal nerves [46].

Specifically, the respiratory and/or olfactory epithelium must be crossed, as trigeminal nerves innervate both epithelia and olfactory nerves innervate only the olfactory epithelium [47,48]. The olfactory epithelium is made up of supporting cells and olfactory sensory neurons consisting of axons that project through the cribiform plate to the olfactory bulb [49].

Generally, molecules are transported from the nasal cavity to the brain via three pathways, intracellular, extracellular, and transcellular, as shown in Figure 1 [50].

#### 2.2.1. Intracellular Transport

The intracellular pathway starts with the internalization of the molecule via endocytosis or pinocytosis by olfactory nerves, as seen in Figure 2, and the cargo-containing endosome is transported through the soma and down the axon to the olfactory bulb [50]. 

A fraction of the molecules moves towards the Golgi and is moved by antero- or retrograde axonal intracellular nerves up to the olfactory bulb [53,54]. The vesicle containing the molecule is moved by the nerves toward the axonal terminal to the olfactory bulb located in the cerebrum. The contents of the vesicle are exocytosed from the olfactory nerve terminal onto the post-synaptic cells of the olfactory bulb [55]. The molecule enters directly into the subarachnoid CSF located around the olfactory bulb and is distributed across the ensheathing astrocytic end-feet into the interstitial fluid surrounding neurons in the brain [56]. Transportation and distribution across the ensheathing astrocytic end-feet is highly dependent and inversely related to molecular weight [56]. A fraction of the molecules in the CSF may enter the brain via Virchow–Robin spaces (perivascular spaces) which surround base membrane capillaries and arteries found in the pial surface [56].

Mitral and tuft cells of the olfactory bulb project to several locations on the ventrolateral surface of the brain [57]. Mitral cells project to different locations viz. amygdala, piriform plexus, entorhinal plexus, and hypothalamus, and tuft cells project to the olfactory nucleus of the cerebrum [57]. These cells ensure the distribution of molecules to different regions of the brain.

Endocytosis has also been observed with trigeminal nerves as seen in Figure 3 and the absorbed molecule is transported directly to the brain [58]. The nerve projections innervate the respiratory region via three branches; ophthalmic, maxillary, and mandibular nerves [59]. Ophthalmic and maxillary nerves pass through the nasal mucosa and extend up to the caudal and rostral parts of the brain near the olfactory bulb via lacerated foramen and the cribiform plate [52,60]. Post endocytosis, an API, may therefore be directly transported to the CNS via axons of trigeminal nerves.

Although both pathways deliver API directly to the brain, some evidence suggests that olfactory nerves may be more efficient [62]. Balin et al. [63] reported that wheat-germ agglutinin conjugated to horseradish peroxidase used as a tracer may be found in the axons of olfactory nerves and in the olfactory bulb after administration onto the respiratory and olfactory regions. Thus, this suggests the possibility of the passage of molecules into the CNS primarily via axons of olfactory nerves.

Axonal transport is a relatively slow process (hours to days) and experimental data reveal rapid rates of API movement; therefore, the complete movement of an API to the brain cannot be attributed to this transport [47,64]. The rapid rate of API transport suggests that axonal transportation is secondary to extracellular transport, which is a more rapid (several minutes) process [50].

The intracellular pathway involves active transportation of the API without limitations of molecular size and lipophilic nature [50]. Hence, this pathway is the primary and predominant mechanism of direct-to-brain delivery from the nasal cavity.

#### 2.2.2. Extracellular Transport

In extracellular transport, the molecule initially passes through the paracellular space across the nasal epithelium and navigates the tight junctions to the lamina propria [65]. The molecule is then transported via bulk flow processes externally along the first and second trigeminal neuronal axon and the channels near olfactory nerves [50]. The axon extends up to the CNS and the molecule is distributed throughout the brain via interstitial fluid movement without primarily passing via the subarachnoid CSF [48,50]. This transport exists due to leakiness caused by co-existence of mature and newly formed neurons on the olfactory epithelium due to slow regeneration of olfactory neurons [66]. Bulk flow of CSF into the brain in combination with the leakiness of neurons enables direct delivery of the API to the brain [66]. This route is mainly for the transport of hydrophilic substances (<1000 Da) viz. proteins and peptides allowing for fast onset of action [67].

#### 2.2.3. Transcellular Transport

The transcellular transport of a molecule occurs across basal epithelial cells via concentration gradients, vesicular, or receptor mediated transport [67]. The molecule crosses to the lamina propia and enters the brain through passive diffusion or active transport [67]. Passive diffusion is responsible for small compounds (<300 Da) and large lipophilic compounds (300–1000 Da) [68,69]. The rate of migration of a molecule is directly proportional to its lipophilicity [70]. This route is mainly responsible for the transport of lipophilic molecules [67].

Generally, the bioavailability of each API after intranasal administration depends on its lipophilic nature; for nasal epithelium permeability, this includes the molecular size and the pathway of CNS delivery.

#### 2.2.4. Enzymatic Activity

Nasal tissue is an effective toxicological barrier presenting with xenobiotic-metabolizing enzymes (XME) largely located on the olfactory neuro-epithelium [52,71]. This region presents with a large network of XME viz. phase I enzymes, which expose or add polar groups onto xenobiotics, and phase II enzymes which conjugate these xenobiotics with several hydrophilic compounds including glutathione and glucuronic acid [72]. The network also consists of transporter proteins that excrete metabolites from nuclear receptors and cells [73].

#### 2.2.5. Elimination

The epithelium of the nasal cavity is covered by a layer of mucosa comprising mucus and cilia responsible mucociliary clearance (MCC) as the primary defense mechanism [67]. During MCC, inhaled particles and pathogens are initially trapped in mucus and pushed by cilia movement to the nasopharynx to clear the nasal cavity of the substances [51]. MCC occurs approximately after every 15 to 20 min [51]. Briefly, goblet cells produce mucous consisting of 90 to 95% water and 5 to 10% of ions, albumin, glycoprotein, immunoglobulin, enzymes, and lysozymes [74]. The foreign material is trapped in the mucous and is moved by cilia towards the nasopharynx facilitated by cilia beat frequency to be swallowed and removed via the digestive system by enzymes and the acidic environment [75,76]. After absorption and transportation to the CNS, relative to its half-life and clearance, the API is transported to systemic circulation and is eliminated via the renal route [77]. In the case of a high dose at the target site, the API is cleared by CSF turnover [78].

#### 2.2.6. Factors Affecting API Transport

Several factors may influence transportation of an API to the CNS, including disease-related alteration, MCC, molecular weight and size, pH of formulation, lipophilicity, isotonicity, API solubility, and access volume.

##### Nasal pH

Nasal physiological pH is generally 4.5–6.5 [79]. The pH of the formulation may affect degree of ionization and API stability and irritate nasal mucosa [67]. 

##### Isotonicity

Formulations that are hypertonic or hypotonic may interfere with cilia movement, thus affecting API absorption [67].

##### Deposition and Access Volume

The administration device, administered volume, and head position during administration may affect extent and localization of API deposition in the nasal cavity [80]. The optimal administrable volume generally retains up to 200 μL for a human nasal cavity, thereby limiting nasal formulations to potent API [80]. The supine position has been to be shown to increase probability of API to reach the olfactory region [67]. The API distribution pattern in the nasal cavity is influenced by the delivery device used for its administration [81]. Ideally the deposited particles should present an aerodynamic particle size of 10 to 20 μm in order to resist intranasal airflow [81,82].

## 3. Bilayered Lipid Nanocarriers

### 3.1. Liposomes

Liposomes are considered the pioneers of lipid vesicular drug delivery systems. They were first described by the British hematologist, Bangham, in 1961 [83]. Liposomes are simple microscopic vesicles in which lipid bilayer structures are present with a central aqueous core volume entirely enclosed by a membrane composed of lipid molecules [84]. 

Liposomes are largely composed of phospholipids and cholesterol [85,86] and offer several advantages over conventional drug delivery systems, such as biodegradability and general non-toxicity. 

Generally, liposomal preparation involves the dissolution of cholesterol, lecithin, and, in some instances, charge inducing agent(s) in an organic solvent and drying to form a thin film. Thereafter, the film is dispersed at a critical hydrating temperature in an aqueous medium to produce a liposome suspension [87]. The hydrating temperature used to prepare liposomes should be above the phase transition temperature of the phospholipid used; that is, the temperature at which transition from a gel to liquid phase occurs, and this can be altered using phospholipid mixtures or by addition of sterols such as cholesterol. Gel state vesicular delivery system is improved by the addition of cholesterol to the lipid phase in the case of liposomes or to the surfactant phase in the case of niosomes [88]. The hydration temperature can provide insight as to the potential for vesicular delivery, system stability, and permeability.

Liposomes have been used for ophthalmic delivery [89], for exploration of the oral administration of insulin [86,90,91], and to enhance the effect and reduce the toxicity therapeutic agents including anti-cancer compounds such as doxorubicin, paclitaxel, vinorelbine, and daunorubicin [92,93]. They have also been explored in the field of infectious diseases for the potential applications in tuberculosis [94,95,96].

Liposomes offer many advantages over conventional drug delivery systems in terms of flexibility for drug delivery and the capability to entrap hydrophilic and hydrophobic drugs [87]. The structure of liposomes may be uni- or multilamellar and can accommodate API in the bilayer structures. The surfaces of the vesicle structure can be functionalized for targeted delivery and therapeutic performance may be enhanced due to reduced clearance from the biological system [97,98]. 

### 3.2. Pharmacosomes

Pharmacosomes are lipid based delivery systems that are colloidal dispersions of drugs that are complexed to lipids [99]. They are termed pharmacosomes due to the linking of a drug (pharmakon) to a carrier (soma). The amphiphilic complexes of phospholipids carry an active hydrogen atom. Depending on the chemical structure of the API–lipid complex, pharmacosomes may exist as ultrafine vesicular, micellar, or hexagonal aggregates [99,100,101]. They are perceived to be more stable than liposomes due to their covalent bond between the API and phospholipid [99].

Pharmacosomes offer numerous advantages over conventional drug delivery systems in terms of flexibility for drug delivery and the capability to entrap hydrophilic and hydrophobic drugs [102]. Pharmacosomes entrap solutes and permit the delivery of API of different solubility via the oral, parenteral, and topical routes of administration. The structure of pharmacosomes may be uni- or multilamellar and can accommodate API in the bilayer structures [99]. The surfaces of the vesicle structure can be functionalized for targeted delivery and therapeutic performance may be enhanced due to reduced clearance from the biological system. Unlike other lipid-based vesicular systems, pharmacosomes have a predetermined entrapment efficiency, as the drug and carrier should possess a stoichiometrically defined unit that is covalently linked [102]. In addition, they offer an advantage of not requiring the removal of unbound (free) API [102]. Due to the presence of a covalent bond, the fluidity of the bilayer is inconsequential on the release rate of the API from the technology. Rather, the release rate is, in turn, dependent on the phase-transition temperature of the API–lipid complex [99,102,103].

The presence of a free carboxyl group or an active hydrogen atom, viz. -NH_2_, -OH, or -COOH on an API can be esterified, with or without a spacer chain, to the hydroxyl group of lipid molecules thereby producing an amphiphilic prodrug. As these newly formed prodrugs are amphiphiles, dilution with water results in the formation of pharmacosomes [103].

Pharmacosomes can facilitate membrane, tissue, or cell wall transfer properties in organisms. The amphiphilic nature of the carriers reduces interfacial tension and, at higher concentrations, is linked to mesomorphic behavior [104]. This decrease in interfacial tension leads to an increase in contact area between the API and the carrier, thereby increasing bioavailability of drugs.

Pharmacosomes have been used with different degrees of success, but a present challenge in the formulation is that the payload must be covalently bound to lipid molecules prior to encapsulation [85,99]. Diclofenac and aceclofenac have both been successfully encapsulated in pharmacosomes, resulting in improved drug delivery efficiency [105,106]. Improved solubility, a reduced side effect profile, and flexible dosing schedules are possible when compared to conventional medicines [100,101,107]. 

### 3.3. Ethosomes and Transferosomes

Transferosomes and ethosomes were created as alternatives to liposomes and niosomes following the partial failure of the latter drug carriers for the delivery of compounds across the skin [108,109,110,111]. Much like their predecessors, transferosomes and ethosomes possess an infrastructure consisting of hydrophilic and hydrophobic moieties, and as such can accommodate drug molecules with a wide range of solubilities [107,111]. They possess high deformability which gives them the ability to penetrate pores with significantly small diameters. Much like the pioneering liposomes, they are made from natural phospholipids and are therefore biodegradable and biocompatible [104,109,112]

They are ultra-deformable and flexible carriers that can facilitate transport between the cells in the skin to reach deeper sites within the skin structure. Ethosomes are flexible due to the incorporation of ethanolic acid and/or ethanolic soya phosphatidyl choline moieties into the bilayer structure. Transfersomes are similar to ethosomes; however, the former use edge activators to impart flexibility into the bilayer [85]. Commonly used edge activators include surfactants such as sodium cholate, sodium deoxycholate, Span^®^ 80, and Tween^®^ 80 [110,113].

These vesicles are capable of transporting molecules that are too large to diffuse through the skin or the systemic route so as to ensure that therapeutically meaningful amounts of macromolecules such as insulin or interferon are delivered [114]. Other applications include the transport of small molecular weight drugs with physicochemical properties that would limit or prevent diffusion across barriers. Transferosomes and ethosomes can be used to target peripheral subcutaneous tissues. For instance, ketoprofen, a non-steroidal anti-inflammatory drug (NSAID), is a commercially available transferosome formulation sold under the trade name Diractin^®^ [112].

Phospholipid inclusion has been attempted in transferosomes and non-ionic transferosomes have been manufactured to deliver piroxicam and interleukin-2 [115,116]. Transferosomes have different mechanisms of penetration in comparison to niosomes and liposomes, and tend to be located in the more hydrated layers of the skin tissues, making them more efficient for transdermal drug delivery [108,115,117,118].

### 3.4. Niosomes

Niosomes are microscopic lamellar structures that are similar to liposomes but are produced using non-ionic surfactants [119]. Niosomes are structurally similar to liposomes, with the main difference being that niosomes are made up of non-ionic surface active agents whereas liposomes include phospholipid compounds [120]. The amphiphiles involved when producing niosomes include sorbitan esters and/or polysorbates and/or ethoxylated alcohols and/or sucrose esters that are stabilized by the addition of cholesterol and small amounts of anionic surfactant such as diacetyl phosphate [100,121,122]

Niosomes, like liposomes, can entrap both lipophilic and hydrophilic compounds in either the aqueous layer or the vesicular membrane that is primarily constituted of lipid materials. Reports suggest that niosomes are more stable than liposomes [108], and the presence of non-ionic surfactants facilitates prolonged circulation of the carriers, potentially enhancing therapy and targeting delivery to the brain, liver, and tumors [123,124].

A schematic representation of the available bilayered nanocarriers is given in Figure 4.

### 3.5. Unilayered Lipid Drug Delivery Systems

#### 3.5.1. Solid Lipid Nanoparticles

Solid lipid nanoparticles (SLN) were developed and introduced at the beginning of the 1990s as a substitute colloidal carrier system for controlled drug delivery [125]. SLN are usually manufactured using lipids that are solid at room and body temperatures, and are stabilized using one or more surfactant(s) [126]. The lipids used can be highly purified triglycerides, mixtures of complex glycerides, or waxes [126]. Due to their lipid-based nature, their popularity has developed rapidly due to biocompatibility and biodegradability, formulation flexibility, and the ability to use well-characterized, generally regarded as safe excipients. In addition, the possibility of controlled drug release and targeting, increased stability, increased loading capacity, avoidance of organic solvent use, and limited issues with scale-up production and sterilization has also facilitated their use [127,128]. SLN have been manipulated to combine the advantages of using other colloidal systems whilst avoiding associated disadvantages of those technologies.

The morphology of SLN is dependent on the method used for incorporation of API into the nanoparticles [128,129]. The incorporation of an API into SLN is reliant on a number of factors, including the solubility of the API in the lipid, miscibility of the API and lipid melt, physicochemical properties of solid lipid matrix and the API, and the method of manufacture [130]. Consequently, three different models of drug incorporation into SLN have been described, viz. type I, or the homogenous matrix type, which consists of API that has been molecularly dispersed within the lipid core or amorphous clusters of API that become homogenously dispersed in the lipid core. Type II SLN, or the drug-enriched shell model, is produced when the lipid is allowed to recrystallize prior to the addition of API during the cooling stages of the manufacturing process, leading to phase separation. In contrast to the drug-enriched shell model, SLN type III, or the drug-enriched core model, occurs when the API precipitates prior to lipid recrystallization, leading to the formation of a lipid shell containing a lower dosage of the drug. This state occurs when the concentration of API in the formulation is close to or at saturation solubility in the molten lipid [128,129].

The major drawbacks of SLN such as API leakage during storage, insufficient drug loading, and the high water content of these dispersions led to the development of nanostructured lipid carriers (NLC) [125,129,131,132]. API expulsion in SLN has been attributed to polymorphic transitions of solid lipids [133]. Polymorphism is the ability of a compound to crystallize into more than one discrete crystalline type, and these exhibit diverse interior lattice structures [128,134]. Lipid molecules exhibit polymorphism and can therefore transition into consecutive crystalline forms without any alterations in their internal structure [134]. Pure lipids exist as low energy and highly ordered crystalline compounds which present as the β-polymorphic form [135]. When the lipids are melted during production of SLN, they may recrystallize into high energy and less ordered α- and/or β′ polymorphic forms [126]. Due to the amorphous nature of these transitional compounds, they allow for incorporation of API which is retained in the lipid matrix. During prolonged storage, the lipid matrix of SLN undergoes a reversal of polymorphic modifications leading to a reduction in the amorphous regions of the matrix owing to α- and/or β′ transition to the stable β form [128,129,130,133], ultimately leading to API expulsion.

#### 3.5.2. Nanostructured Lipid Carriers

NLC have similar physicochemical properties to SLN; however, they have been developed by nano-structuring of the architecture of the lipid matrix in order to increase API loading in addition to preventing drug leakage during storage, thereby resulting in greater flexibility for targeting specific API release profiles [125,128]. The structural matrix of NLC is derived from a mixture of solid and liquid lipids that are mixed in specific combinations, resulting in a less ordered lipid matrix with numerous imperfections that can accommodate a greater amount of API. NLC particles therefore solidify on cooling, but do not recrystallize, and remain in the amorphous state [128,129,132,133]. API expulsion on prolonged storage is less likely to occur from NLC as compared to SLN due to the differences in architecture of the lipid matrices.

Type I NLC, or the imperfect type of NLC, consists of a lipid matrix with numerous imperfections that are able to accommodate API molecules. This model is obtained by mixing solid lipids with sufficient liquid lipid(s) to avoid the formation of a highly ordered structure on cooling. The voids created in the structure that accommodate the API molecules are formed as a result of different chain length fatty acids and mixtures of mono-, di-, and triacylglycerols [128,129,136].

Type II NLC, or the amorphous type, are obtained by mixing lipids such as hydroxyoctacosanyl hydroxystearate, isopropyl myristate, or dibutyl adipate that fail to recrystallize following homogenization and cooling of nano-emulsions. These lipids are therefore able to form amorphous NLC that prevent recrystallization of the lipid matrix following cooling, thereby allowing the retention of API within the lipid structure whilst minimizing API expulsion on storage [128,129].

Type III NLC are multiple model systems comparable to multiple emulsions such as w/o/w emulsions. These systems are comprised of minute nano-compartments within the solid lipid matrix that are formed as a result of phase separation [128,129]. These systems are obtained by mixing solid and liquid lipids, such as medium-chain or long-chain triacyl glycerols and oleic acid, in a ratio such that the amount of liquid lipid is greater than that of the solid lipid [125,128,129,137]. At high temperatures, the molten solid lipid and the liquid lipid are miscible. However, on cooling (40 °C) of the nano-emulsion, the solubility of the liquid lipid in the solid lipid is exceeded, that is, a miscibility gap is achieved, and phase separation occurs. This leads to precipitation of the liquid lipid and the formation of tiny droplets in the molten solid lipid. Subsequent solidification of the solid lipid leads to concentration of oily nano-compartments [128,129]. This model has been shown to permit an increased loading capacity for API that have a higher solubility in liquid lipids than in solid lipids [138].

#### 3.5.3. Lipid-Drug Conjugates (LDC)

Lipid drug conjugates (LDC) are active moieties/compounds that are covalently modified with lipids. The lipid modification increases the lipophilicity of the drug and is also responsible for altering other physiochemical properties of the conjugated molecule. The fundamental property of LDC is the ability of the lipid matrix to specifically conjugate with water-soluble/hydrophilic compounds, thus offering a platform for the development of novel pro-drugs, especially in oral drug delivery applications, for various therapeutic strategies [104,139]. Owing to the chemical nature of drugs and lipids, different conjugation strategies and chemical linkers can be applied in the synthesis of LDC. These conjugation strategies/linkers include drug conjugation with fatty acids, steroids, glycerides, and phospholipids, as well as linkage via ester, amide, hydrazine, disulphide, and other bonds such as thioethers [139]. The aforementioned techniques have been shown to be vital in determining drug release from LDC, thus influential in final product performance [139,140]. Nonetheless, although LDC can be individually administered, most of them are loaded into appropriate delivery systems to enhance drug targeting for optimized therapeutics.

#### 3.5.4. Nanocapsules

Lipid nanocapsules (LNC) are nanocarriers of a biomimetic nature with an amalgam structure between polymeric nanoparticles and liposomes. They have a lipidic core composed of medium chain triglycerides surrounded by a surfactant shell made of a PEGylated surfactant and optionally lecithin or other co-surfactants. The choice of excipients that constitute the oily/lipidic phase and the surfactant can be tailor made to suit the properties of the encapsulated molecule [141,142]. LNC are prepared using the phase inversion temperature method, which is a solvent free and low energy technique. Briefly, the components are mixed and the resultant emulsion is then subjected to numerous heating and cooling cycles (between 60 °C and 90 °C) to obtain reversible emulsion phase inversions. Higher temperatures result in w/o emulsions due to the dehydration of the polar surfactant heads while lower temperatures yield o/w emulsions. Subsequent to the various temperature cycles, rapid dilution with cold water is usually performed at a temperature in the phase inversion zone to obtain the final suspension of LNC. Rapid cooling, without dilution, can also be utilized to obtain these nanocarriers [142,143]. 

#### 3.5.5. Micelles

Micelles are aggregates composed of surfactant phospholipids that are dispersed in a liquid [144]. Micelles constitute a hydrophobic core and a hydrophilic outer shell, or vice versa depending on the nature of the dispersion medium, with conventional micelles resulting from small molecules with hydrophilic/polar/charged “heads” and a hydrophobic tail [145]. Generally, the hydrophobic core of the micelle is capable of sequestering or dissolve hydrophobic molecules until they are released by some drug delivery mechanism [144,145]. Diblock, triblock, or more complex structural copolymers with alternating hydrophilic and hydrophobic moieties can also form micelles in aqueous media. Their hydrophilic blocks usually consist of poly(ethylene oxide), while the hydrophobic blocks consist of poly(propylene oxide), poly(lactic acid), or any other biocompatible hydrophobic polyethers or polyesters [145,146]. In addition, hydrophilic polyethylene glycol is applied in the assembly of micelles due to its neutral and nontoxic nature. Other hydrophilic polymers that have been applied in micellar fabrication include poly(N-vinyl pyrrolidone) and pNIPAM [147,148,149].

#### 3.5.6. Self-Nanoemulsifying Drug Delivery Systems

Self-nanoemulsifying drug delivery systems (SNEDDS) are homogenous liquid mixtures consisting of oil, surfactant, a drug, and a co-emulsifier or solubilizer, which spontaneously form an oil-in-water nanoemulsion following aqueous dilution with gentle agitation. In the human body, the required agitation to form nanoemulsions is provided by the digestive motility of the gastrointestinal tract [150]. The oil phase in nanoemulsions may include triglycerides such as tri-, di-, or mono-acylglycerols, vegetable oils, mineral oils, or free fatty acids, with the selection based on the solubility of the drug. The most commonly used surfactants for drug and food delivery include sorbitan fatty acid esters (Spans), polyoxyethylene derivatives of sorbitan fatty acid ester (Tweens), polyoxyl-35 castor oil (Cremophor^®^ EL), lauroyl macro-golglycerides (Gelucire^®^ 44/14), polysaccharides, phospholipids, and amphiphilic proteins. As an ultra-low negative interfacial tension is required for the formation of nanoemulsions, co-surfactants or co-solvents are used in conjunction with a surfactant. Commonly applied co-surfactants or co-solvents include poly ethylene glycol, propylene glycol, ethanol, diethylene glycol monoethyl ether (Transcutol-P/HP), ethylene glycol, glycerine, and propanol [151,152,153,154].

### 3.6. Submicron Emulsions 

#### Microemulsions and Nanoemulsions

An emulsion is defined as a colloidal system comprised of one liquid dispersed as small globules in an immiscible liquid [155]. The small globules are termed the dispersed phase whereas the immiscible liquid is termed the continuous phase. Emulsions are classified into two basic types according to the volume ratio of the liquids, viz. a water-in-oil (w/o) system consisting of water dispersed in an oily continuous phase and an oil-in-water (o/w) system in which oil is dispersed in an aqueous continuous phase [155]. Preparation of an emulsion involves force and addition of an amphiphilic molecule comprised of both hydrophilic and hydrophobic moieties. 

A microemulsion (ME) is defined as a single optically isotropic and thermodynamically stable system consisting of two phases: an aqueous and lipid phase [156,157]. The phases are separated by an interfacial layer of a surfactant(s) and a co-surfactant(s) which aids in achieving ultralow interfacial tension necessary for the formation of globules [158,159]. ME exhibit globule sizes ranging between 10 and 200 nm, below the optical wavelength, resulting in a generally transparent appearance [159,160,161]. 

ME are categorized into three basic types, o/w, w/o, and bicontinuous, according to the ratio of the two phases [162]. An o/w ME is made up of an aqueous continuous medium, oil at an equal or smaller quantity to aqueous phase, surfactant, and a co-surfactant [161]. Oil globules are dispersed in an aqueous continuous phase and provide a suitable environment for the solubilization of lipophilic compounds within the globules. 

A w/o ME consists of a non-polar solvent, water at a lower quantity to that of the non-polar solvent, surfactant, and a short-chain lipophilic alcohol as the co-surfactant [161,163]. The nature and type of the dispersed phase is primarily intended for solubilizing the drug and to ensure its absorption through biological membranes [164]. The surfactant and co-surfactant are also responsible for drug solubilization, in addition to the emulsification capacity of the system [165]. A bicontinuous ME is made up of interconnected sponge-like domains of oil and water separated and subsequently stabilized by localized surfactants existing as surface interfaces [158,163,166]. 

ME are suitable for drug delivery due to their ease of preparation [167], unstable API protection [168], enhanced solubility of lipophilic API [169], controlled release of API [162], improved bioavailability via high absorption rates and enhanced penetration of biological membranes [170], improved shelf-life [171], and reduced intra- and inter-individual variability in the pharmacokinetics of an API [172]. The solubility of hydrophilic and lipophilic API may be improved by solubilization within globules and additional sites at the interfacial film [169,171]. Hence, in an ME system, drug loading is permitted at saturation solubility and thermodynamic activity is enhanced, resulting in the drug favorably partitioning toward and permeating through biological membranes [173].

Nanoemulsions (NE) present as an isotropically transparent blend with similar constituents as ME but a lower content of oil and a higher content of hydrophilic surfactants and co-surfactants [174]. They are kinetically stable and present with smaller globule sizes ranging from 10 to 100 nm [174,175]. Contrary to ME, preparation of NE involves low to high energy methods, viz. ultrasonication [176], high pressure homogenization [177], emulsion inversion point [178], phase-inversion temperature [179], and bubble bursting [180]. NE have shown great potential for application as a drug delivery system, particularly by enhancing the bioavailability of Biopharmaceutical Classification System II and IV drugs [181,182]. This is achieved by the reported bioactive attributes of this system, such as enhancing the solubility of poorly water soluble drugs [183], increasing gastro-intestinal tract membrane absorption of poorly permeable drugs [184], increased intestinal lymphatic pathway absorption [185], circumventing hepatic-first pass metabolism [186], and inhibiting P-gp efflux [187].

The stability of emulsions is primarily determined by the appearance of signs of destabilization, including creaming, cracking, phase sedimentation, sedimentation, coalescence, flocculation, and loss of drug content. ME and NE are generally more stable than conventional emulsions, as reported by Figueiredo et al. [188], where there was gradual loss of API content in conventional emulsions after 30, 60, and 90 days where there was no loss observed in API-loaded ME during the same time period.

The types of unilayered lipid-based nanocarriers are represented in Figure 5.

## 4. The Application of Lipid Based Nanocarriers in the Treatment of Neurological Disorders

### 4.1. Neurodegenerative Disorders

#### 4.1.1. Alzheimer’s Disease

Alzheimer’s disease (AD), a progressive neurodegenerative CNS disorder, is responsible for 60 to 80% of the dementia cases among the elderly [189]. The primary pathological characterization of AD is the presence of intracellular neurofibrillary tangles (NFTs) composed of hyperphosphorylated τ-protein and extracellular deposition of amyloidal-β (Aβ) protein as senile plaques in the hippocampus of the brain, which plays an important role in the regulation of emotional responses [190,191]. The clinical manifestations of AD are associated with cognitive, memory, and behavioral impairments, causing mood fluctuations and delirium, and subsequently leading to dementia [190,191]. 

Currently, AD is untreatable, with a few drugs commercially available that target the Aβ and τ proteins to improve cognitive symptoms and slow down the degeneration process. The first-line drug candidates are acetylcholinesterase (AChE) inhibitors such as donepezil (DPL), rivastigmine (RS), and galantamine (GH) for the management of mild to moderate and severe cases [192]. Memantine, an N-methyl D-aspartate (NMDA) antagonist, is mostly used in patients with moderate to severe AD. It acts by reducing neurotoxicity caused by the over activation of the glutaminergic system, which further prevents neuronal loss [193,194]. The inability of these drugs to cure or modify the disease condition led to the discovery of new drug molecules such as aducanumab, a disease-modifying monoclonal antibody. Aducanumab was approved by the United States Food and Drug Agency (FDA) in 2021 based on the mechanism of reducing the Aβ load. To date, the drug is still considered a new agent, and its clinical efficacy, adverse effects, and cost benefit ratio are still debatable [195]. Approval of this therapy highlights the importance of early detection and accurate diagnosis. A summary of the potential targets for the treatment/alleviation of AD are depicted in Appendix A. 

Although aducanumab reduces amyloid plaque, increased doses result in amyloid related imaging abnormalities of oedema [196,197]. Similar to aducanumab, several other biologics such as Aβ therapeutic antibodies, τ-therapeutic antibodies, triggering receptor expressed on myeloid cells 2 (TREM2) therapeutic antibodies, and nerve growth factors (neurotrophin) are currently under clinical trials. It is approximated that 40% of these studies were discontinued due to lack of clinical significance and only 12% entered Phase III. The lack of clinical significance is mainly attributed to the inability of these antibodies to cross the BBB [198]. 

The use of low molecular weight molecules has been suggested to be a beneficial approach to treat AD due to their molecular properties that enables them to generally cross the BBB. The molecular properties of interest include, but are not limited to, molecular weight of the drug < 400 Da, polar functionality with less than seven hydrogen bonds, and polar surface area <52–80 Å^2^ [199,200]. Approximately 187 of small molecules are under various stages of clinical trials, among which close to 40% have been discontinued and only six (~3%) have been approved for use in the treatment of AD [198]. The main hindrance, however, remains the inability of some of the approved small molecules to cross the BBB [201]. As such, there is a need for using innovative drug delivery techniques that can transport potentially therapeutic moieties across the BBB.

Several drug delivery strategies have been used to improve the permeability of drugs across the BBB. Among these are the use of physical techniques such as microneedles, lasers, electroporation, focused ultrasound, magnetophoresis, microwave, and chemical techniques, viz. inclusion complexes and penetration enhancers capable of opening the tight junctions. In addition, biological techniques such as the use of viral vectors (adeno associated virus) stimuli have been suggested [199,202,203]. However, the use of these strategies on a commercial scale are associated with limitations such as high production costs, access to specialized equipment, patient comfort, risk of infection, and toxicity [202,204]. With the advent of nanotechnology and advanced research in polymer chemistry, new opportunities arise to overcome these limitations. Nanocarriers, specifically those based on lipidic excipients, are proven to possess ideal properties for the delivery of drugs across the BBB. Apart from being bio-degradable, non-toxic, and non-immunogenic [199], their affinity and binding with endothelial cells and ability to adsorb into blood capillaries provide sustained drug delivery and surface modulating abilities with functional proteins and cell penetrating peptides. These make them good candidates for targeted drug delivery systems across the BBB for effective treatment of AD [205,206]. 

DPL, an API approved by the USFDA, is a hydrophilic small molecule for the treatment of AD with limited BBB permeability. Consequently, it is associated with increased dosing frequency and subsequent severe cholinergic side effects. To mitigate this, DPL-loaded SLN were prepared for delivery to the brain via the intranasal route by solvent emulsification diffusion technique using glyceryl monostearate (GMS) as the lipid and a blend of Tween^®^ 80 and poloxamer 188 (1:1) as surfactant [207]. In vivo studies conducted in male albino Wistar rats by administering DPL-SNL via the intranasal route and DPL solution (DPL-sol) intravenously showed better brain target delivery, with significantly high concentration of drug in the brain following intranasal administration. During stability studies, DP-SLN were found to be stable at 4 ± 2 °C and 25 ± 2 °C/60 ± 5% RH for 6 months, thus indicating the potential for commercial applications [207].

In another study by Sood et al., curcumin and donepezil co-encapsulated NLC were prepared by microemulsion technique and administered via the intranasal route [208]. The in vivo pharmacokinetic studies showed higher drug concentration in the brain for NLC, compared to the free drug administered intravenously [208]. NLC also resulted in significant increase in acetylcholine (AChE) and reduced the levels of oxidative stress. In addition, the Wistar rats used for in vivo applications of the study showed improved memory and learning compared to animals that received the free drug [208].

GH, an AChE inhibitor, is reported to have similar drawback as DPL. GH-loaded SLN were prepared by microemulsification using glyceryl behenate (Compritol^®^) as the lipid, Pluronic^®^ F-127 as surfactant, and Tween^®^ 80 as co-surfactant. GH-SLN were evaluated for in vitro drug release and in vivo memory restoration capability in cognitive deficit adult Wistar rats [209]. GH-SLN showed a prolonged drug release over 24 h with 99% of the drug released as compared to pure GH with 81% of the drug released in the first 1 h. Pharmacokinetic parameters observed after oral administration indicated a 2-fold increase in bioavailability of the drug from GH-SLN [209]. The in vivo pharmacodynamics studies conducted using the Morris water maze test showed a significant reduction in the mean escape latency for the GH and GH-SLN treated groups in comparison with the isoproterenol induced cognitive deficit group, indicating the effectiveness of the SLN approach. The study also indicated good stability for GH-SLN at 4 ± 2 °C and 25 ± 2 °C storage conditions [209].

Various scientists have extensively studied RS lipid nanoparticles to understand the penetration of the drug across the BBB. Among all the approved AChE inhibitors, RS has the lowest bioavailability (~40%) and very short half-life (t_1/2_) of approximately 2 h, resulting in negative effects on the treatment outcomes [210]. Liposomes of RS, when combined with a cell-penetrating peptide, resulted in a significant increase in the concentration of the drug in CNS cerebral tissues over the normal liposomes and the free drug [211]. RS liposomes administered intranasally [211], subcutaneously [212], orally, and intraperitoneally [213] in various animal models resulted in increased penetration of the drug into the brain and higher inhibition of AChE when compared to the free drug.

Presently, most of the research focus is on developing disease modifying drugs (antibodies for Aβ and τ proteins) and small molecules that can address the oxidative stress and neuroinflammation. However, the challenge of BBB permeation remains, with observations in the aforementioned studies indicating the potential for lipid based nanocarriers to circumvent these challenges in the treatment of AD. In addition, administration via the intranasal route presents an additional promising opportunity to circumvent many of the difficulties associated with API permeability across the BBB.

The combination of these therapeutically active compounds with intranasally administered nanolipid-based drug delivery systems could be the way to improve the success rate of new molecules for better therapeutic outcomes.

#### 4.1.2. Parkinson’s Disease

Parkinson’s disease (PD), also known as paralysis agitans, is an age-related neurodegenerative disorder, affecting 2 to 4% of the population over the age of 50 years and second only to AD in its debilitating nature [214,215]. PD is a complex multi-system condition, associated with motor, non-motor, and neuropsychiatric manifestations. Motor symptoms include tremor, rigidity, postural abnormalities, stooped posture, bradykinesia, akinesia, and festinating gait. Non-motor symptoms include olfactory problems, constipation, sleep dysfunction, and dysautonomia, while neuropsychiatric problems include dementia and cognitive impairment that has a great impact on the quality of life of the patient [216,217,218]. Pathologically, PD is mainly characterized as the loss of melanin-containing dopaminergic neurons of the substantia nigra compacta of the mid brain, resulting in a decrease in dopamine content in the striatum and accumulation of α-synuclein (α-SN) protein aggregates [219,220]. Research shows that mitochondrial dysfunction, oxidative stress, and genetic factors also play an important role in the etiology of the disease [221].

Similar to AD, PD is also untreatable at present and no treatment is available to either stop or slow down the progression of the disease. Treatment options available mainly relieve the symptoms and assist in improving the quality of life of the patient [222]. Levadopa is the primary drug of choice in the treatment of motor related symptoms, followed by the use of dopamine agonists, catechol-O-methyltransferase inhibitors (COMT), or monoamine oxidase-B (MAO-B) inhibitors as adjuvants to inhibit the metabolism of dopamine in patients with advanced PD. The non-motor symptoms are treated using tricyclic antidepressants, selective serotonin reuptake inhibitors, and selective serotonin norepinephrine reuptake inhibitors for depression, as well as CNS stimulants for fatigue, AChE inhibitors for dementia, and antipsychotics for psychosis [223]. A summary of the drug targets is depicted in Appendix A.

In an effort to overcome the medical limitations associated with current PD drugs such as levodopa-induced dyskinesias, a number of new compounds that can modify disease progression are currently in development or in the early phases of clinical trials. The development mainly includes the use of antibody immunotherapy to reduce α-synuclein protein aggregates, improve mitochondrial function, and reduce neuroinflammation by reducing oxidative stress, as well as drugs with multi-functional properties that can address both motor and non-motor symptoms, and gene and stem cell-based therapies to replace the lost dopaminergic neurons [224]. The clinical activity of the current treatment options, the newly developed molecules, and novel therapeutics as solo agents has been hindered by their short half-life, in vivo stability, and limited concentration in CNS due to difficulties with crossing the BBB, among other factors [225]. The use of lipid based nanocarriers, especially liposomes, exosomes, nanoemulsions (NE), and SLN, have been extensively investigated by various researchers to overcome the above-mentioned challenges.

Over the past two decades, the SLN of several dopamine agonists were reported in the literature for their in vitro and in vivo performance. SLNs of piribedil [226], bromocriptine [227], apomorphine, and ropinirole hydrochloride have been previously discussed [203,223,225]. These reports highlighted the role of SLN in improving bioavailability [228,229] and prolonging drug release that resulted in stable plasmatic drug levels and prolonged action [227]. Similarly, ropinirole hydrochloride-loaded SLN exhibited a maximum therapeutic index at a decreased dose and dosing frequency in comparison with the commercial formulation [230].

The enzymes GTP cyclohydroxylase-1 (GCH1), tyrosine hydroxylase (TH), and aromatic acid decarboxylase (AADC) are primarily responsible for the synthesis of dopamine as well as the conversion of L-Dopa administered in patients with PD [231]. The loss of dopaminergic nigrostriatal neurons that express AADC results in reduced synthesis and conversion, leading to erratic response to L-Dopa therapy. A promising approach to this problem seems to be introducing these genes to regulate the dopamine synthesis apparatus. In addition, significant research is underway to use growth factors (GF), particularly glial cell-line derived neurotrophic factor (GDNF), neurturin (NRTN), artemin (ARTN), and persephin (PSPN), cerebral dopamine neurotrophic factor (CDNF), mesencephalic astrocyte-derived neural factor (MANF), brain-derived neurotrophic factor (BDNF), and vascular endothelial growth factor (VEGF) for the treatment of PD [231]. To date, numerous preclinical tests have been conducted, and the use of GF seems promising in disease modification by protecting as well as promoting the regeneration of nigrostriatal dopamine neurons [232]. The one major limitation with GF is their ability to cross the BBB and poor distribution in the tissues, leading to the use of surgery or high infusion volumes to attain therapeutically effective concentrations in CNS. The use of high concentrations/volumes can lead to vector induced overexpression of α-synuclein in the brain [233], leading to the use of novel drug delivery systems such as nanocarriers for the delivery of gene therapy.

In a recent study, phospholipid-based gelatin nanoparticles encapsulating basic fibroblast growth factor (bFGF) were prepared for intranasal administration in hemi-parkinsonian rats. The gelatin nanostructured lipid carriers were prepared with nonionic copolymer-poloxamer 188 and solid lipids using the water-in-water emulsion method and then freeze-dried [234]. Increased levels of bFGF in the striatum and the olfactory bulb were attributed to poloxamer 188 induced transcellular transport, which also acted as a penetration enhancer by interacting with the lipid bilayer of the nasal mucosa. In addition, the strong surface negative charge of the carriers reduced the impact of mucociliary clearance of the drug and increased the residence time, leading to increased bioavailability. Though the study did not find significant therapeutic outcomes in terms of neuro protective effect of the GF, it has indicated the potential for using nanocarriers as potential drug delivery systems for delivery of the genes across BBB [234].

In another study, chitosan (CS) coated NLC were prepared by encapsulating GDNF using the melt emulsification technique [235]. The surface of the particles was modified using transactivator of transcription (TAT) peptide, a cell penetrating peptide (CPP) to obtain CS-NLC-TAT-GDNF. The in vivo performance of the carriers was evaluated by intranasal administration in a 1-methyl-4-phenyl-1,2,3,6- tetrahydropyridine mouse model of PD [235]. The CS-NLC-TAT-GDNF-treated group revealed motor recovery at two weeks when compared to the control group [235]. The promising results from these studies give much hope for further developments which can pave the path towards clinical trials in human models.

Various literature reports have alluded to the role of lipid-based antioxidant nanocarriers capable of reducing the neuroinflammation associated with oxidative stress [203,223,225,236]. Gold NP of lipoic acid [237], SLN of idebenone (IDE) [238], NE of combined vitamin E and coenzyme Q10 [239], liposomes and NE of resveratrol [240,241], and NE selegiline [242] have been studied. The results from in vitro and in vivo studies of these nanocarriers were found to be promising in addressing bioavailability issues associated with short half-life, extensive first pass hepatic metabolism, and poor penetration across the BBB [203,223,225,236,237,238,239]. Several of these studies highlighted the benefits of using the intranasal route of administration and its great potential for achieving desired therapeutic outcomes in the treatment of PD.

Most recently, selegiline lipid nanocapsules (LNC) loaded with quercetin, a P-glycoprotein (P-gp) pump inhibitor, were developed for oral administration [243]. SEL was found to be poorly absorbed orally due to extensive first pass metabolism and sensitivity towards P-gp pumps. LNC were formulated using a modified emulsion-sonication method. Permeability studies conducted using everted gut sac showed a 4-fold and 6-fold increase in the permeation of SEL from SEL-QUR LNC in comparison with SEL-QUR and SEL suspensions, respectively. Confocal laser scanning microscopy (CLSM) pictures of intestine showed a 2-fold deep penetration of SEL with SEL-QUR LNC [243].

Mursaleen et al. manufactured micellar nanocarriers of hydroxytyrosol (HT) using a modified thin-film hydration method [244]. HT is well known for its antioxidant and neuroprotective properties; however, it has limited absorption across the BBB [245,246]. The results from the research highlighted enhanced mean BBB passage of HT by 50%. This resulted in increased protection against rotenone induced cytotoxicity and oxidative stress by up to 12% and 9%, respectively [244].

Vitexin is a flavonoid derived from *Crataegus pinnatifida Bunge*. Vitexin-loaded SLN were manufactured and evaluated for their effect in a 6- hydroxydopamine (6-OHDA) induced PD animal model [247]. The vitexin-loaded SLN therapeutics in 6-OHDA induced Parkinson’s clearly showed improvement in total reactive antioxidant activity in mice [247]. It was also observed that ROS in the striatum were diminished by vitexin-loaded SLN. Furthermore, the said formulation was efficient in averting memory destruction in Morris water maze test as well as depressive-like behavior in tail suspension test [247].

#### 4.1.3. Dementia

Neurodegenerative diseases such as AD and PD result in damage or injury to brain cells, in turn leading to dementia. Degenerating neurites near amyloid plaques have been fingered to be associated with developing dementia in patients with AD [248,249]. In particular, the global population with dementia has been reported to be approximately 24 million, with 60% of these cases being due to AD, arising in at least 1% of those between 50 to 70 years of age and approximately 50% for those over 70 years [250,251]. Owing to this, numerous novel therapeutic options have focused on managing the progression of AD (§ 2.1.1), thus preventing the occurrence of dementia. Nonetheless, certain studies that have reported on the novel application of certain lipid nanocarriers in the treatment or management of dementia are highlighted in this section.

Yadav et al. [252] demonstrated that SLN loaded with resveratrol could be a promising therapeutic strategy in the management of vascular dementia. In this study, the authors evaluated the therapeutic effects of the aforementioned nanocarriers in a permanent bilateral common carotid artery occlusion (BCCAO) induced model of vascular dementia. The SLN formulation showed a 4.5 higher drug delivery level to the brain compared to the free drug. Neurobehavioral analyses revealed that the nanocarriers ameliorated cognitive decline observed in BCCAO rats. Furthermore, BCCAO animals administered with drug-loaded SLN showed a significant reduction in the generation of mitochondrial reactive oxygen species, lipid peroxidation, and protein carbonyls. Moreover, the lipid carriers significantly improved the redox ratio and the activity of Mn-superoxide dismutase, as well as facilitating a significant reduction in hypoxia-inducible factor 1α levels, while the levels of nuclear factor (erythroid-derived 2)-like 2 and heme oxygenase 1 increased [252].

Prathipati et al. [253] investigated the neuroprotective effect of curcumin, a molecule with limited bioavailability, on homocysteine induced oxidative stress in vascular dementia. In this study, the authors prepared curcumin-loaded SLN using the solvent evaporation technique prior to administering the carriers to male Sprague Dawley rats. Following grouping of the animals (curcumin + homocysteine treated, normal saline, and homocysteine groups), behavioral assessments were performed on days 0, 7, and 14 with subsequent exsanguination for biochemical, neurochemical, and histopathological assays in different regions of the brain. The authors reported that treatment with a high dose of curcumin-loaded SLN (curcumin—SLN −25 mg/kg, p.o + homocysteine −400 µg/kg/day, i.v) successfully ameliorated the oxidative stress in vascular dementia following elucidation of observed biochemical, neurochemical, and histological alterations. This suggests that lipid encapsulated molecule is a potential therapeutic option for the management of oxidative stress in vascular dementia induced by homocysteine in different brain regions [253].

The incorporation of didanosine into NLC for potential treatment of AIDS dementia complex was reported [254]. In this study, the authors manufactured aqueous didanosine-free and didanosine-loaded NLC using hot high-pressure homogenization, and subsequently used the preferential in vitro adsorption of apolipoprotein E (Apo E) onto the surface of colloidal drug carriers as a strategy to evaluate the in vivo potential of the carriers to transport didanosine to the brain using two-dimensional polyacrylamide gel electrophoresis. Following their investigations, the authors concluded that lipid carriers stabilized using Solutol^®^ HS 15 alone or with a ternary surfactant system consisting of Solutol^®^ HS 15, Tween^®^ 80, and Lutrol^®^ F68, were capable of preferentially adsorbing Apo E, thus revealing that didanosine-loaded NLC may have the potential to deliver the antiretroviral to the brain in vivo [255].

A Box–Behnken Design optimized NLC-loaded transdermal formulation of rivastigmine for the treatment of dementia was investigated. In vitro release studies showed a more sustained drug release performance from the NLC-loaded transdermal patches in comparison to the commercially available Exelon^®^ (rivastigmine, RS) patch. Furthermore, pharmacokinetic studies showed an increased C_max_ and AUC_0–72_ in plasma from Albino Wistar rats treated with NLC-loaded transdermal patches when compared to the marketed Exelon^®^ patch. From their findings, the authors suggested the RS NLC based transdermal patch as a potential carrier for improving bioavailability of the compound for the better treatment and management of dementia [256].

Zidovudine, a nucleoside reverse transcriptase inhibitor used in antiretroviral therapy, has been reported to possess poor CNS penetration because it is a substrate of the active efflux transporter which is readily expressed in the brain. Owing to this, subtherapeutic levels delivered to the CNS lead to ineffective treatment of conditions such as AIDS dementia complex and HIV encephalopathy [257]. Consequently, Dalpiaz et al. investigated conjugation of ursodeoxycholic acid with zidovudine to form a lipid drug conjugate (prodrug) encapsulated in poly lactic-co-glycolic acid (PLGA) in the presence of Pluronic-F68 as a surfactant that would evade the AET systems. Overall, the prodrug was reported to show significant safety and efficacy compared to earlier mentioned zidovudine prodrugs because it demonstrated the potential for CNS delivery [258].

Cai et al. proposed the use of hollow manganese Prussian white nanocapsules to attenuate τ-related pathology and alleviate cognitive decline via simultaneously easing neuroinflammation, scavenging reactive oxygen species, and reducing hyperphosphorylated τ-proteins, thus offering a treatment strategy for τ-related neurodegeneration diseases, such as AD and frontotemporal dementia [259]. In this study, the authors constructed the self-assembly hollow manganese Prussian white nanocapsules via a facile biomimetic mineralization and the carriers showed good ROS scavenging activity to relieve oxidative stress due to their variable valence states and low oxidation-reduction potentials. In addition, the carriers inhibited the phosphorylation of τ-protein via the AKT/GSK-3β pathway and reduced neuroinflammation via suppressing the activation of microglia and astrocytes, as well as reducing inflammatory cytokine release. Moreover, the nanocapsules attenuated τ-related pathology and cognitive deficits in Okadaic acid-induced Alzheimer’s disease-like rats via relieving oxidative stress, counteracting neuroinflammation, and inhibiting the phosphorylation of Tau without significant side effects [259].

Polysialic acid, a hydrophilic and endogenous carbohydrate polymer, based micelles were designed to deliver a calmodulin antagonist for the treatment of vascular dementia. In this study, polysialic acid was initially chemically conjugated with octadecylamine to form a polysialic acid– octadecylamine copolymer that subsequently self-assembled into micelles in aqueous solution. The calmodulin antagonist-loaded micelles showed sustained drug release over 72 h and could cross the BBB mainly via active endocytosis by brain endothelial cells followed by transcytosis. In addition, the calmodulin antagonist-loaded micelles significantly reduced the escape latencies of right unilateral common carotid arteries occlusion mice compared to free drug following a water maze test for spatial learning. Furthermore, the decreased hippocampal phospho-CaMKII (Thr286/287) and phospho-synapsin I (Ser603) was partially restored as well as an elevation in BrdU/NeuN double-positive cells was observed in the study animals after treatment with the antagonist-loaded micelles. The developed micelles were postulated to have great potential for treatment of vascular dementia and possibly promote research in improving the efficacy of brain-targeted drug delivery [260].

Symptomatic treatment of dementia has been reported to involve the targeting of cyclooxygenase (COX) enzymes mediated neuroinflammation, which is primarily a consequence of using nonsteroidal anti-inflammatory drugs. Inhibition of COX enzymes can reduce amyloid deposition and inhibit glia activity, and has been shown to provide symptomatic relief in a mouse model of AD [261]. Consequently, Sipos et al. developed meloxicam-loaded polymeric micelles, using a Quality by Design approach, for potential nose-to-brain delivery in the treatment of neuroinflammation. The authors reported a more than 20 times quicker dissolution rate of meloxicam and five-fold increase in nasal permeability compared to the free drug, thus postulating the profound potential for in vivo brain distribution of the micellar system as a novel approach to effectively treat neuroinflammation [262].

### 4.2. Epilepsy

Epilepsy is a chronic neurological disorder that presents as recurrent seizures caused by excessive cellular activity due to membrane depolarization and subsequent release of excitatory neurotransmitters in some parts of the brain [263,264]. The seizures are of variable intensity and duration from person-to-person and may include co-existing conditions [80]. Drug therapy for epilepsy is directed at long-term controlling of symptoms and the current anti-epileptic drugs are administered orally or intravenously. The oral route presents with several challenges for the bioavailability of drugs, viz. gastro-intestinal tract (GIT) fluid dissolution, GIT permeability, hepatic first pass metabolism, protein-binding, and crossing the blood-brain barrier. Similar challenges are presented with the intravenous route, despite those of the GIT, and additionally this route is invasive. Hence, an alternative route to target the brain may be beneficial to the efficacy of the drug and improving patient compliance and/or adherence. The mechanism of action and targets for anti-epileptics is depicted in Appendix A.

In recent years, microemulsions (ME) and nanoemulsions (NE) have been investigated for use as potential nanocarrier systems for the delivery of antiepileptic drugs, more especially for direct-to-brain delivery via the intranasal route. The sub-micron diameter of the globules is preferred, as smaller particles increase the surface area, elicit rapid biological membrane permeation, and enhance brain uptake [265,266].

The solubility of a drug serves great importance to its bioavailability as dissolution is a prerequisite step to absorption through biological membranes. ME were investigated by Porecha et al. [267] to increase the solubility of carbamazepine (CBZ), a poorly water-soluble anti-epileptic drug. The CBZ-loaded ME consisted of oleic acid, Tween^®^ 80, Transcutol^®^ as the oil, surfactant, and co-surfactant, respectively, and were intended for intranasal administration for direct-to-brain delivery to reduce systemic side effects and increase brain uptake [267]. The manufactured ME were administered to two populations of Sprague Dawley rats. One group of rats received a dose of CBZ before seizure induction to monitor control, and the other group was dosed after seizure induction to assess treatment. Antiepileptic activity was assessed in different stages, viz. the time elapsed until convulsions occur after seizure induction (S1) and recovery time after seizure induction (S2). The ME were compared with a CBZ solution, and S1 was shorter after administration of the ME, revealing their potential for enhancing the management and control of epilepsy. The onset of the neuroprotective activity of CBZ in a nanocarrier system was also enhanced, as shown by a shorter S2. The solubility of CBZ was successfully increased in an aqueous based ME and diffusion through sheep nasal mucosa was faster due to the non-polar components of the formulation, as these act as permeation enhancers. ME provided an alternative for direct-to-brain delivery via the intranasal route as the efficacy of the antiepileptic drug was enhanced.

In another study, pharmacodynamic evaluation was conducted after oral or intranasal administration of CBZ-loaded ME consisting of oleic acid, Tween^®^ 80, and propylene glycol as the oil, surfactant, and co-surfactant, respectively [268]. CBZ delivered from the ME via the intranasal route exhibited a significant reduction in the duration of a seizure, extension phase, clonus phase, and stupor phase of convulsions in seizure-induced rats. The higher efficacy of the CBZ-loaded ME may indicate the delivery of a more effective dose via the intranasal route as systemic circulation and the BBB are avoided [268].

During an epileptic attack, P-gp activity is over-expressed in capillary endothelial cells of the BBB and transport substrates are removed from the CNS via P-gp efflux mechanisms [269]. Removal of drugs via efflux reduces their bioavailability in the CNS. Several anti-epileptic drugs are P-gp substrates and may play a role in drug resistance [269]. Surfactants have been reported to be P-gp inhibitors with the extent of inhibition depending on the hydrophilic–lipophilic balance and the critical micelle concentration [270].

Topiramate, used for the treatment of primary generalized tonic–clonic seizures and partial onset seizures, is a P-gp substrate, and in order to prevent its removal from the CNS, Patel and Parikh [271] manufactured o/w NE. The NE consisted of Capmul^®^ MCM C8, Tween^®^ 20, and Carbitol™ as the oil, surfactant, and co-surfactant, respectively. In vivo studies were performed with albino Wister rats to assess the absorption of topiramate into the brain and its efficacy in reducing or treating induced febrile seizures. The NE formulation consisted of 500 mg topiramate per 10 mL and was intranasally or orally administered to each rat during a seizure to assess the time of recovery from the convulsions. The NE was suitable for intranasal administration as it exhibited low viscosity (6.97 cP) and pH (5.25) comparable to that of nasal fluid. The recommended viscosity for formulations intended for intranasal administration is < 500 cP [272]. Average seizure duration after intranasal administration was shorter compared to after oral administration, indicating a faster onset of action with the former route due to direct-to-brain drug delivery. This suggests that the BBB was circumvented after intranasal administration via the olfactory or trigeminal pathways found on the respiratory and olfactory regions of the nasal cavity [47,50,273]. The efficacy of the NE was also compared with a topiramate-loaded suspension and the former was found to treat convulsions in a shorter duration, suggesting the inhibition of P-gp efflux by the NE. The superiority of NE was further evidenced as a greater quantity of topiramate was found in the brain 1 h after intranasal or oral administration compared to the suspension, indicating increased brain uptake. In addition to the nanomeric size of the globules, brain uptake via the intranasal route may be enhanced by the surfactant, as it may increase permeability by reducing the interfacial tension on the olfactory region [274].

Figueiredo et al. [188] manufactured phenobarbital (PB)-loaded ME intended for oral administration for the treatment of epilepsy. PB was incorporated at two proportions, 5% and 10%, consisting of Labrasol^®^, ethanol, isopropyl myristate, and purified water as the oil, surfactant, co-surfactant, and aqueous phase, respectively. For comparative studies, a PB-loaded conventional emulsion was also manufactured. The nanomeric size of the dispersed globules ranged from 10 to 300 nm and were influenced by the content of PB incorporated in a formulation [188]. As the content of PB increased in each formulation, the average diameter of globules increased, likely due to localization of PB in the interfacial layer of each globule. The PB-loaded ME presented an average polydispersity index below 0.5, indicating homogeneity of the globule size. Homogeneity is a critical factor, as it may affect the dosing accuracy and thereby compromise the efficacy and safety of a product. The in vitro release kinetics were assessed and PB was reported to be released from the ME 4.8 times higher than in the emulsion after 24 h. Hence, ME show potential to increase drug release in a shorter time, thereby shortening the onset of action [188].

Abbas et al. [275] conducted research to deliver and clonazepam (CZ) intranasally. Superparamagnetic iron oxide (SPION87)-loaded NLC were prepared to deliver clonazepam to the brain. The technology was further combined with an in situ thermoresponsive mucoadhesive gelling system. The technology was investigated for its brain targeting ability through the trigeminal nerve and olfactory bulb pathway [275]. The optimized formulation SPION-87 NLC were loaded into the thermoresponsive in situ gel prepared with Pluronic^®^ F-127 as temperature triggered gelling agent and sodium alginate as an anionic mucoadhesive agent [275]. The results of this study suggested that the use of SPION-NLC enhanced the anticonvulsant activity of CZ in albino rats. However, it must be noted that the brain targeting in this study was achieved by the application of an external magnetic field [275].

Similarly, embelin, a benzoquinone derivative that exerts its anticonvulsant effect by improving GABAergic transmission and reducing dopamine levels in the CNS, was loaded in NLC with the intention of improved brain targeting [276]. The embelin-loaded NLC that was manufactured by use of the emulsification method accompanied by ultrasonication consisted of cetyl palmitate as solid lipid, octyldodecanol as liquid lipid, and Poloxamer 188 as the surfactant [276]. The pharmacokinetic assessment revealed that intranasal administration of embelin-loaded NLC significantly improved the pharmacokinetic parameters when compared to the plain API and marketed formulations. Summarily, the study demonstrated improved overall bioavailability and brain targeting potential of API-loaded NLC after intranasal administration [276].

Alam et al. [277] formulated lamotrigine (LMT)-loaded NLC with the intention of delivering the API to the brain via the intranasal route. The solvent evaporation techniques was used to develop LMT-loaded NLC with GMS, oleic acid, and a mixture of Poloxamer 188 and Tween^®^ 80 used as the solid lipid, liquid lipid, and the surfactant mix, respectively [277]. The prepared NLC was optimized using a Box–Behnken Design (BBD). The ex vivo drug permeation experiments performed on excised goat nasal mucosa showed a substantial increase in the nasal permeability of LMT-loaded NLC which is likely due to increased mucoadhesivity imparted by the Poloxamer 188 [277].

Valporic acid (VA), an API used in both epilepsy and bipolar disease, was loaded in NLC for intranasal delivery [278]. The VA-loaded NLC were prepared by emulsion solvent diffusion-evaporation method. Cetyl palmitate and octyldodecanol were used as the solid lipid and liquid lipid, respectively. The stabilizer and surfactant used in these experiments were soy lecithin and Poloxamer 188, respectively. The in vivo study on male Wistar rats showed significantly higher brain:plasma concentration upon intranasal administration of VA-NLC than the control intranasal and intraperitoneal dose. These data suggest that the NLC formulation provides better anti-epileptic efficacy [278].

Additional advantages of using of lipid nanocarriers for direct-to-brain via the intranasal route include non-invasiveness, painless, and ease of application, making these nanocarriers patient-friendly.

### 4.3. Ischaemic Stroke

An ischemic stroke is an acute neurologic dysfunction of vascular origin, in which blood supply to the brain is interrupted and hemorrhagic. This interruption results in imbalances in pH, glucose, oxygen, and other nutrients, leading to the death of brain cells. At present, the only approved pharmacological treatment available for this condition is recombinant tissue plasminogen activator, which has thrombolytic activity towards blood clots, thus restoring normal blood flow [279]. The mechanisms of action of some API used to treat ischaemic stroke and offer neuroprotection are depicted in Appendix A.

Ischemic strokes produce disproportionate free radicals, which induce apoptotic cell death of neurons. Consequently, specific brain targeting activity of baicalein, a flavonoid derived from the root of *Scutellaria baicalensis* Georgi, loaded NLC following intravenous injection was investigated. The reported NLC formulations comprised tripalmitin, Gelucires^®^, vitamin E, phospholipids, and poloxamer 188. Following intravenous administration of an aqueous control solution containing baicalein as well as the baicalein-loaded nanocarriers in male Wistar albino rats, baicalein plasma levels from the latter were much higher and the half-life much longer than those from animals in the control group. Furthermore, a 7.5- versus a 4.7-fold baicalein accumulation was observed in the cerebral cortex and brain stem, respectively, of the NLC-treated animals compared to those that received the aqueous solution. In line with these findings, higher baicalein accumulations following NLC administration were also observed in the hippocampus, striatum, thalamus, and olfactory tracts of the animals. Overall, the lipid carriers improved the ability of baicalein to penetrate the brain; therefore, the NLC showed potential as a promising lipid-based drug delivery system for the treatment of CNS conditions such as ischemic stroke [280]. The effect of baicalin-loaded PEGylated cationic SLN modified with OX26 antibody on regulating the levels of baicalin and amino acids during cerebral ischemia–reperfusion in rats was reported. In addition, the authors investigated the effect of these nanocarriers on excitotoxic neuronal injury as well as determining the pharmacokinetics of baicalin in cerebrospinal fluid during an ischemia–reperfusion period. The AUC of baicalein from formulated NLC was found to be 5.69-fold higher than that of the baicalein solution, while the *C*_max_ of the encapsulated molecule was 6.84-fold higher than that of the solution, thus confirming the potentially better therapeutic efficacy of the lipid carriers. Following the determination of extracellular levels of aspartic acid, glutamate, taurine, glycine, and γ-aminobutyric acid to deduce the imbalance in amino acids caused by ischemia and reperfusion, the lipidic carriers showed a superior effect than the baicalin solution in reducing aspartic and glutamic acid while increasing glycine, taurine, and γ-aminobutyric acid levels during an ischemia–reperfusion period. The demonstrated improved bioavailability of baicalin in cerebral spinal fluid of rats with cerebral ischemia–reperfusion injury, as well as the better protective effect against cerebral ischemia injury, show the potential of these carriers for use in brain targeted therapy [281].

Lu et al. [282] demonstrated that PEGylated-lipid nanoparticles conjugated to a Fas ligand antibody were detected in an ischemic region of the brain during therapeutic targeting. In this study, 3-n-butylphthalide, a constituent in celery oil, was encapsulated in the conjugated lipid nanoparticles and successfully delivered to the ipsilateral region of an ischemic brain of male C57BL/6J wild-type mice. The encapsulated nano-drug delivery system showed significant improvements in brain injury and neurological deficit following ischemia while using a significantly lower dose than pure 3-n-butylphthalide [282].

Vinpocetine, a derivative of the vinca minor alkaloid vincamine, is well-documented for use in the treatment of ischemic stroke and other cerebrovascular disease [283]. However, its clinical application is largely hindered by being sparingly water soluble, having a low and irregular dissolution rate in the GIT due to pH effects, and a high first pass-effect, emanating in a low oral bioavailability of approximately 7% [284]. Consequently, Lin et al. developed vinpocetine-β-cyclodextrin-tartaric acid-loaded NLC using the emulsification ultrasonic dispersion method. The pharmacokinetic in vivo behavior of the formulation was determined using New Zealand rabbits, with the NLC formulation showing a 592% relative bioavailability compared with a pure suspension of vinpocetine [285]. Similarly, Morsi et al. prepared vinpocetine SLN, with Tween 80 and Pluronic^®^ F 68 as the surfactant mixture, using a modified high shear homogenization method followed by an ultrasonication technique for use as a brain targeted sustained drug-delivery system. The cumulative percentage drug release of the optimal formulation was 72.12% after 96 h, with zero-order release kinetics and a controlled release profile, thus displaying the potential to deliver vinpocetine to the brain [286].

The protective effects of indomethacin-loaded nanocapsules against oxygen-glucose deprivation in organotypic hippocampal slice cultures was investigated in an in vitro model of ischemia using organotypic hippocampal cultures exposed to oxygen-glucose deprivation. Following treatment of oxygen-glucose deprived cultures with indomethacin nanocapsules, the cell death was observed to significantly reduce. In addition, treatment with the nanocapsules evidently repressed the levels of pro-inflammatory cytokines (IL-1β and IL-6) as well as TNF-α levels. Treatment with 100 μM of indomethacin-loaded nanocapsules reduced the levels of ERK1/2 and JNK phosphorylation to those of control cultures, as well as promoting the activation of inducible nitric oxide synthase (iNOS). Moreover, glial activation induced by oxygen-glucose deprivation was prevented following treatment with the drug-loaded nanocapsules. These findings demonstrated indomethacin-loaded nanocapsules as a promising neuroprotective therapeutic agent for the management of cerebral ischemia, most likely through inhibiting inflammatory cascades [287]. Liu et al. reported the development of microRNA nanocapsules for improved brain delivery to disease sites of the brain. Following the administration of intravenous encapsulated microRNA-21, an endogenous miRNA with well documented angiogenic, anti-apoptotic, and regenerative effects in the treatment of cerebral ischemia into a rat model of cerebral ischemia, the authors observed improvements in the infarct volume, neurological deficit, and histopathological severity [288]. Jin et al. [289] encapsulated intravenous immunoglobulin into 2-methacryloyloxyethyl phosphorylcholine nanocapsules to optimize therapy by facilitating BBB penetration for the selective targeting of the immunoglobulin to ischemic regions. The nanocapsules were found to effectively cross the BBB with site-specific accumulated delivery of intravenous immunoglobulin in a high-affinity choline transporter 1-overexpression dependent manner via endothelial cells in brain ischemic areas. Overall, the delivery system was deemed a potential approach to effectively deliver drugs to ischemic regions of brain tissue at low therapeutic doses [289].

Edaravone is used to save ischemic brain tissues by eradicating excess reactive oxygen species (ROS) without the limitation of a therapeutic time-window (thrombolytic agents are administered approximately 4.5 h following onset of symptoms to avoid the increasing risk of intracerebral hemorrhage). However, its therapeutic efficacy is dampened by its short half-life and inadequate permeation across the BBB. Owing to this, Jin et al. formulated edaravone-encapsulated agonistic micelles for delivery of the molecule into the brain for the management of ischemia. Their findings showed that the agonistic micelles could actively up-regulate endothelial monolayer permeability in vitro. In addition, more of the drug was observed in ischemic brain tissue following micellar treatment compared with pure edaravone after intravenous injection in a mouse model, with the micelles being observed to be superior to the free drug at salvaging ischemic tissue. Moreover, the highest efficiency in accelerating axonal remodeling in the ipsilesional white matter and improving functional behaviors of ischemic stroke models was observed following treatment with the developed micelles [290]. A rapamycin micelle system, consisting of ROS-responsive and fibrin-binding polymers, was developed to concurrently regulate numerous cell types involved in oxidative stress, neuroinflammation, and brain vascular impairment, which occur after ischemic stroke onset thus causing formation of microthrombus formation and resulting in neuronal death. In vivo results from an ischemic stroke rat model showed an entire remodeling of the investigated neurovascular unit following micelle polarized M2 microglia repair and preservation of the BBB, leading to improved neuroprotection and blood perfusion [291]. The feasibility of applying the benefits of mRNA in preventing neuronal death in a rat model of transient global ischemia was investigated. An mRNA encoding brain-derived neurotrophic factor was encapsulated into a polyplex nanomicelle, with the lipidic formulation showing a significant increase in the survival rate of hippocampal neurons after transient global ischemia. Furthermore, a Y-maze behavioral test confirmed enhanced spatial memory in treated rats compared with untreated rats 20 days after initiation. The reported findings also pointed to astrocytes as the major targets of the mRNA-loaded nanomicelles, implying that the increased secretion of brain-derived neurotrophic factor from astrocytes provides a microenvironment that enables neurons to endure changes caused by ischemic stresses, as well as terminating progressive neuronal death occurrence following an ischemic attack [292].

Singh et al. [293] developed and optimized a liposomal delivery system for an ethyl acetate fraction of *Allium cepa* outer scales, to investigate its neuroprotective efficacy in a mouse model of cerebral ischemia–reperfusion injury. The liposomal formulation was prepared using the thin film hydration method and their neuroprotective effects, viz. cognitive/sensorimotor functions, reduction in cerebral infarct size, and brain oxidative stress of the optimized carrier system was determined after induction of cerebral injury in Swiss Albino mice. Intra-nasal administration of the optimized liposomal formulation showed significant neuroprotection in the mice at 1/10th the oral dose following grouping of the experimental animals (Sham control; oral vehicle; oral drug; nasal blank liposomes, nasal drug-loaded liposomes) [293]. Moreover, Bruch et al. provided a comprehensive review on the use of liposomes for drug delivery in stroke, highlighting the pioneering or seminal studies in the field as well as fairly recent studies in the last 29 years [294].

### 4.4. CNS Neoplastic Disease

The treatment of primary brain and CNS tumors constitutes a considerable and daunting aspect of clinical practice. In an attempt to deliver chemotherapeutics to brain related tumors, many strategies have been proposed and/or used to circumvent these shortcomings. Some of these include local delivery of chemotherapeutics and other treatments, either directly into the tumor bed at the time of initial resection, or via implantable catheters, pumps, or slow-release polymers that can provide more stable pharmacokinetics. In addition, therapy via the intrathecal route using lumbar punctures or implanted Ommaya reservoirs can be used. This has been particularly applied in treatment of CNS lymphomas and leptomeningeal carcinomatosis [295].

The use of convection-enhanced drug delivery may result in an increase in the effective target volume for many pump-based delivery systems [296]. Systemically, chemo-osmotic methods of temporarily disrupting the BBB to improve pharmacokinetics of intravenously administered agents have also been attempted, primarily in treating gliomas. Nevertheless, drug delivery across the BBB or directly into tumor or parenchyma by any of these “forced” methods were met with limited clinical success [296]. As such, the treatment of such aggressive neoplastic diseases requires a technology that systemically delivers drugs across the BBB.

SLN have been applied for the delivery of chemotherapeutic agents to the brain. Some examples include temozolomide [297], etoposide [298], and paclitaxel [299]. When formulated as an SLN, etoposide exhibited more effective inhibition of the proliferation of glioma cell lines when compared to conventional etoposide alone, and was accompanied by a concomitant decrease in cytotoxicity in astrocytes [298]. This ultimately resulted in a widening of the therapeutic window. Similarly, SLN-loaded paclitaxel treatment resulted in inhibition of glioma cell growth in vitro by several orders of magnitude higher than observed in non-entrapped therapy [299].

SLN have been used to deliver camptothecin to the brain as both proof of concept and a potential treatment of brain neoplasms [300,301]. The research conducted demonstrated enhanced accumulation, distribution, and retention of camptothecin-loaded SLN in the animal brain. In addition, there was evidence of superiority of in vitro anti-tumor activity of camptothecin-loaded SLN when compared with camptothecin in solution/suspension or in physical mixture with SLN [301]. Similarly, paclitaxel and doxorubicin-loaded SLN were developed, exhibiting improved physicochemical as well as BBB crossing ability when compared to solutions and conventional delivery mechanisms [302,303,304].

Lipid nanocapsules have also been successfully used to treat CNS related tumors, specifically gliomas. Ferrocenyl diphenol tamoxifen derivative, a highly active novel organometallic compound, as well as paclitaxel have been loaded into lipid nanocapsules and effectively used to reduce the occurrence of multidrug resistance as well as produce positive outcomes [299,305].

Saha et al. [306] designed and developed novel β-amphetaminylated cationic lipid nanoparticles for the treatment of glioblastoma multiforme. The NP were found to be non-cytotoxic and were able to permeate through the BBB. The mechanism was presumed to be through active transcytosis [306]. Furthermore, the lipid nanoparticle was capable of co-encapsulating paclitaxel and PDL1-siRNA. The co-encapsulate-loaded lipid nanoparticle technology exhibited apoptosis driven cellular cytotoxicity against GL261 cells and improved the overall survivability of orthotopic glioblastoma bearing mice compared to the non-targeting technology [306].

Emami et al. [307] developed a transferrin conjugated paclitaxel (PTX)-loaded NLC for treatment of brain cancer. The research group used the emulsification-solvent evaporation technique utilizing cholesterol, triolein, and Poloxamer 188 as the solid lipid, liquid lipid, and surfactant, respectively [307]. Furthermore, stearyl amine was used as the amino group bearing group anchor for transferrin. Transferrin was covalently bound with the surface of PTX-NLC. Formulation optimization was performed by use of a BBD. The in vitro cytotoxicity study performed via MTT assay on the U-87 cancer cell found that the transferrin coupled PTX-NLC slightly increased the cell viability when compared to the marketed formulation (Anzatax^®^), while there was a concentration-dependent tumor cell inhibition. While drug conjugation with transferrin modified NLC sustained the drug release and provided a biocompatible and biodegradable drug delivery system, in vivo performance of the technology will need to be investigated [307].

The API used for the treatment of brain cancer, temozolomide (TMZ) was encapsulated in NLC by use of high-pressure homogenization to improve the outcomes of treatment [308]. Gelucire^®^ and Vitamin E were used as solid and liquid lipid, respectively, while Tween^®^ 80 and Transcutol^®^ in a 6:4 ratio were used as a surfactant. In addition, mannitol was added to the final formulation as cryoprotectant [308]. Response surface methodology (RSM), specifically BBD, was used to optimize the TMZ-loaded NLC with regard to entrapment efficiency, particle size, polydispersity index, and ζ-potential. The TMZ-NLC were further assessed for their pharmacokinetic profile [308]. The loaded NLC demonstrated a substantial increase in in vivo brain distribution and pharmacokinetic investigation after intranasal administration. This was observed for TMZ-NLC when compared to both the intranasal drug solution and the intravenous formulation. The intranasally administered NLC enhanced the brain permeability, biodistribution, C_max_, T_max_, and area under the curve (AUC) [308].

Song et al. studied NLC modified with arginine-glycine-aspartic acid peptide (RGD) for brain targeting of TMZ [309]. They utilized solvent diffusion method for NLC preparation by using Compritol^®^ 888 ATO, Cremophor^®^ ELP, and soya phosphatidylcholine (SPC) as solid and liquid lipids and surfactant, respectively. The prepared RGD-TMZ-NLC was further evaluated for physicochemical properties, in vitro release, cytotoxicity and in vivo therapeutic profile. The in vitro drug release profile of the NLC demonstrated sustained drug release from the NLC, which became more persistent after coating with RGD [309]. The targeted NLC also increased the cell inhibition/cytotoxic effect of the drug with an increase of ~10 times than TMZ solution and ~2 fold than TMZ-NLC. The in vivo study demonstrated similar results to the in vivo study. Greater tumor inhibition ~83.3% was observed for RGD-TMZ-NLC demonstrating a promising nanocarrier system for glioma therapy.

In other similar work, Qu et al. also attempted to deliver TMZ to the brain [310]. TMZ-loaded SLN and NLC were developed alongside PLGA polymeric nanoparticles with the intention of estimating the aptness of nanocarrier system towards the treatment of glioblastoma multiforme. The TMZ-loaded NLC was formulated by solvent diffusion method making use of Compritol^®^ 888 ATO, Cremophor^®^ ELP, and Tween^®^ 80 as solid lipid, liquid lipid and surfactant, respectively. The study demonstrated the use of this technology to achieve sustained drug release. In the in vitro cytotoxicity assay, it was illustrated that the NLC was the most efficient carrier among the developed nanocarriers with regard to reduction in viability of glioma cells. Similarly, the results of the in vivo estimation of anti-tumor efficiency was identical to the in vitro data confirming that NLC was associated with higher tumor regression [310]. Overall, the study shows NLC has better potency to target brain tumor cells than the other carrier systems [310].

In attempt to treat meningeal leukemia, Sharma et al. developed Tween^®^ 80-coated NLC for brain targeting of cytarabine (CYT) [311], an API commonly used to treat metastatic and meningeal leukemia in the CNS. However, it is associated with limited therapeutic efficacy due to lower permeability across the BBB, drug resistance, and cytotoxicity to the healthy cells [312,313]. The NLC were formulated using stearic acid, oleic acid, and polysorbate 80 as solid lipid, liquid lipid, and surfactant, respectively. The fabrication of NLC was accomplished using. In this work, the in vitro MTT assay and cytotoxicity assay performed on EL-4 cells demonstrated an initial higher cytotoxic effect of CYT-loaded NLC when compared to the CYT solution. However, this decreases over time owing to the sustained release properties of the CYT-loaded NLC [311]. However, the blank NLC did not exhibit any cytotoxic effect, suggesting that the technology does not modify the cytotoxic response of the drug but instead modulates the drug release. Moreover, the NLC demonstrated sufficient stability in varying atmospheric conditions [311].

Wu et al. also utilized the versatility of lipid nanocarriers to achieve brain targeting by co-loading TMZ and vincristine (VNC) in both SLN and NLC [314]. The SLN were developed by solvent displacement method with stearic acid and soy lecithin as lipid phase, while the NLC was formulated by solvent diffusion method using Compritol^®^ 888 ATO as solid lipid and Cremophor^®^ ELP as the liquid lipid while using SPC as the stabilizer. Both the SLN and NLC were designed to possess a positive ζ-potential to facilitate the drug carrier system towards the negatively charged tumor cells. The cytotoxicity assay on U87MG cells demonstrated significantly higher concentration dependent tumor cell inhibition by the co-loaded NLC than SLN. Similarly, the in vivo anti-tumor study also demonstrated that the co-loaded NLC more promptly inhibited the growth of tumor tissues than the co-loaded SLN and individual drug-loaded NLC [314].

Along the same line of thought, Chen et al. combined TMZ with gene therapy to enhance the chemotherapeutic efficacy of TMZ [315]. The NLC was prepared using Compritol^®^ 888 ATO, Cremophore^®^ ELP, and a combination of SPC and Tween^®^ 80 utilizing the solvent diffusion technique to co-deliver TMZ and green fluorescent protein plasmid (DNA) [315]. The in vitro cytotoxicity assay performed using the U87 MG cell line demonstrated significant cell growth inhibition. The in vivo cell assay demonstrated similar anti-tumor efficacy on the mice model. Overall, the co-delivery of TMZ-DNA-loaded in NLC demonstrated a higher tumor inhibition ability than the single-drug solution, while also showing superior brain tumor targeting [315].

A summary of the aforementioned nanocarriers and their applications in various CNS disorders are summarized in Table 1.

## 5. Conclusions and Future Perspectives

Advances in medicines and therapeutics over the years have led to research into targeted drug delivery for potential clinical translation for the management of neurodegenerative disorders. Furthermore, the advancement in pharmaceutical nanotechnology in the design and manufacture of nano-structured particles has shown their potential applications in the delivery of pharmaceutical agents to different target organs that are otherwise inaccessible using conventional approaches, such as the CNS, which is protected by the presence of the BBB. Herein, nano-lipid drug delivery systems for potential use in treating some common diseases affecting the CNS, including Alzheimer’s, Parkinson’s, and dementia, have been presented as having favorable attributes such as prolonged drug release, increased CNS bioavailability, and more improved therapeutic efficacy than corresponding free drug solutions/suspensions.

Although lipid nanocarriers have greatly advanced research in CNS targeted delivery over the years, to date, none of these nanocarriers have been granted market authorization following the successful completion of all stages of clinical trials. While the aforementioned benefits of using lipid-based carriers underscores the need to fast-track their translational development into clinical practice, technological advances need to be initiated to achieve appropriate capacity for scalability and the production of cost-effective affordable dosage forms. Nonetheless, following the ongoing technological advancements in biotechnology, including the application of monoclonal antibodies and mRNA as therapeutic agents, the latter being applied in the development of some currently used COVID-19 vaccines, we anticipate the development of associated strategies that will harness the advantages of the aforementioned nanocarriers for treating various neurological disorders in the foreseeable future.

## Figures and Tables

**Figure 1 pharmaceutics-14-00836-f001:**
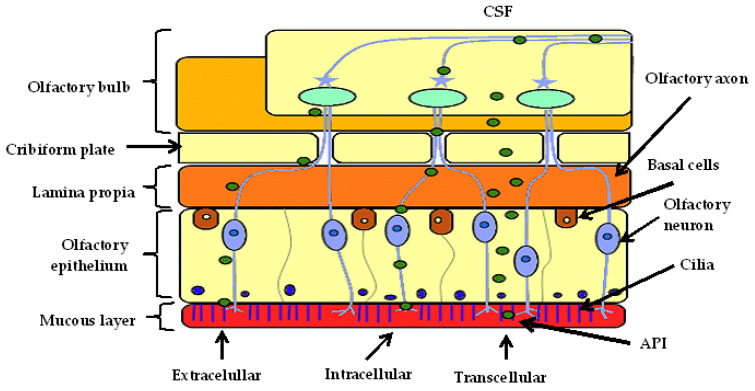
Schematic representation of potential mechanisms of direct-to-brain delivery of API via the nasal cavity (Adapted with permission from [51]. Copyright 2022, copyright holder Elsevier, Amsterdam).

**Figure 2 pharmaceutics-14-00836-f002:**
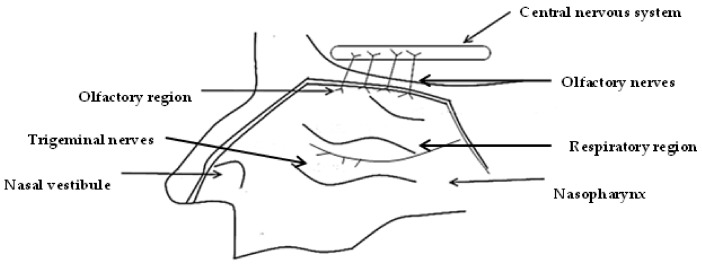
Image of the different areas of the nasal cavity (Adapted with permission from [52]. Copyright 2022, copyright holder Elsevier, Amsterdam).

**Figure 3 pharmaceutics-14-00836-f003:**
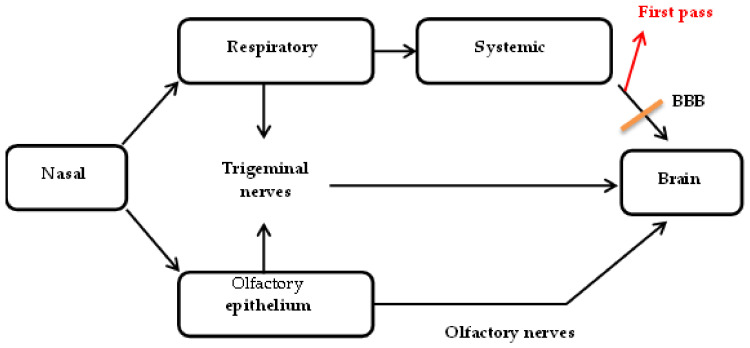
Potential transport of API from the nasal cavity to the brain (Adapted with permission from [61]. Copyright 2022, copyright holder Elsevier, Amsterdam).

**Figure 4 pharmaceutics-14-00836-f004:**
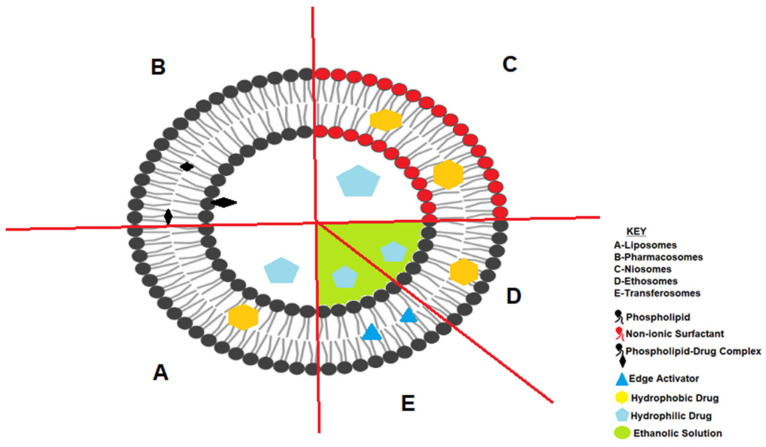
Schematic representation of the various bilayered lipid–lipid based nanocarriers. Liposomes (**A**); Pharmacosomes (**B**); Niosomes (**C**); Ethosomes (**D**); and Transferosomes (**E**).

**Figure 5 pharmaceutics-14-00836-f005:**
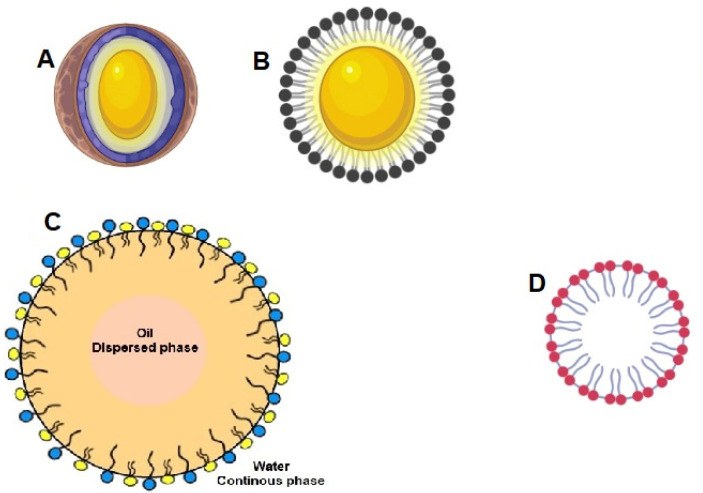
Schematic representation of the various unilayered lipid-based nanocarriers. Nanocapsules (**A**); SLN/NLC (**B**); Nanoemulsion (**C**); and Micelle (**D**).

**Table 1 pharmaceutics-14-00836-t001:** Summary of Nanocarrier applications in CNS disorders.

Disease	API	Nanocarrier Type	Lipid System	Technique	RoA	In Vivo Model	In Vivo Results	Refs
Alzheimer’s	Donepezil	SLN	Glyceryl monostearate	Solvent Emulsification–Diffusion Technique	IN	Wistar Rats	Improved bio distribution and pharmacokinetics	[207]
	Donepezil + Curcumin	NLC	Precirol (Solid lipid) and Capmul MCM (liquid lipid)	Hot High-Pressure Homogenization	IN	Wistar Rats	Higher drug concentration in the brain	[208]
	Galantamine	SLN	Glyceryl behnate (Compritol^®^)	Micro-Emulsification	IN	Wistar Rats	Significant memory restoration capability and two-fold increase in bioavailability	[209]
	Rivastigmine	Liposomes	Egg phosphatidylcholine and Cholesterol	Ammonium Sulphate Gradient Loading Method.	IN SC PO and IP	Male Sprague Dawley rats	Enhanced BBB penetration and improved brain delivery	[211,212,213]
	Rivastigmine	Liposomes	Phosphocholine, dihexadecyl phosphate and cholesterol	Lipid Layer Hydration and The Heating Methods	SC	Wistar Rats	Preventing amyloid plaque formation	[211,212,213]
	Rivastigmine	Liposomes	Dipalmitoylphosfotidyl choline (DPPC) and cholesterol	Film Formation Method	IP and PO	Balb-C type mice	Inhibition of AChE	[211,212,213]
Parkinson’s	Ropinirole HCL	SLN	Dynasan 114 and stearylamine	Emulsification-Solvent Diffusion	IN	Mice	Reduction in signs of Parkinsonism-like symptoms and comparable with marketed oral formulation	[230]
	Piribedil	SLN	palmitic acid (PA) and polyvinyl alcohol (PVA)	Hot Homogenization Followed by Ultrasonication	IN	Wistar rats	Four-fold increase in AUC, increased nose-to-brain uptake	[226]
	Bromocriptine	NLC	Tristearin, stearic triglyceride (tristearin), Miglyol 812, caprylic/capric triglycerides (tricaprin) Mivaplex 600, stearic monoglyceride (monostearin)	Homogenization Or Ultrasonication	IV	Sprague–Dawley rats	Rapid onset of action, long lasting activity (5 h)	[227]
	Apomorphine hydrochloride	SLNs	tripalmitin, Hydrogenated soybean phosphatidylcholine	Emulsification	PO	Wistar rats	A 12- to 13- fold higher bioavailability	[229]
	Basic fibroblast growth factor	Phospholipid-based gelatin nanoparticles	N/A	Water-In-Water Emulsion and Freeze-Drying Technique	IN	hemiparkinsonian Sprague–Dawley rats	Stimulated dopaminergic function	[234]
	Glial cell-derived neurotrophic factor	TAT- LNC	(Precirol ATO and Mygliol^®^	Melt Emulsification Technique	IN	male C57BL/6J mice	Reduction in microgliosis and motor recovery	[235]
	Selegiline	NE	grape seed oil and Sefsol 218	High Energy Emulsification Method	IN	Wistar rats	Decreased dopamine loss, Brain: blood ratio of 2.207 indicating the ability to deliver the drug into brain	[242]
	Selegiline and quercetin	LNC	Labrafil M 2130 CS and Capryol 90	Modified Emulsiosonication Method	IP	Wistar rats	Increased behavioral response and effect of quercetin on increasing the bioavailability by modulating the P-gp inhibitor	[243]
	Vitexin	SLN	N/A	Hot Melt Emulsification And Ultrasonication Method	N/A	Mice	Neuroprotective effect observed via improved total reactive antioxidant in striatum	[247]
Dementia	Resveratrol	SLN	stearic acid, lecithin and taurocholate	Refer To Method	PO	Sprague Dawley (SD) rats	Reduced mitochondrial oxidative stress	[252]
	Aphanamixis Polystachya leaf extract (ethanol)	Liposomes	Phospholipid extracted from Egg yolk	Solvent Injection Method	PO	Swiss albino mice	Significant improvement in memory function, locomotor activity, and ambulatory performance	[249]
	Curcumin	SLN	D-L-Homocysteine, Glyceryl monostearate, and Stearic acid	Modified Solvent Evaporation Method	caudal vein	Sprague Dawley rats	Reduced oxidative stress	[253]
	Rivastigmine	NLC	Combination of solid and liquid lipids	High Pressure Homogenization Technique	TD	Wistar rats	Increased Cmax and AUC, resulting in increased bioavailability	[256]
Epilepsy	Carbamazepine	ME	oleic acid	Oil- In- Water Emulsion	IN	Sprague Dawley rats.	Significantly higher CBZ ME via intranasal route and protection from seizures	[267]
	Topiramate	NE	Capmul MCM C8	Oil- In- Water Emulsion	IN and PO	Wistar rats	Improved bioavailability	[271]
	Clonazepam	SLN and NLC	Glycerol oleate and oleic acid	High Pressure Homogenization Technique	IN	Swiss Albino mice	NLCs showed prolonged the onset times for convulsion	[275]
	Embelin	NLC	solid lipid (cetyl palmitate) and liquid lipid (octyldodecanol)	Emulsification And Ultrasonication Method	IN	Wistar rats	Higher concentration of drug into brain for NLCs as compared to plain embelin and marketed formulation	[276]
	Lamotrigine	NLC	Glyceryl monostearate as solid lipid and oleic acid as liquid lipid a	Solvent Evaporation Method	PO and IN	Wistar rats	Accumulation and longer retention of the drug in the brain Significant improvement in latency and duration of tonic hindlimb extension, up to 24 h post seizure induction.	[277]
	Valproic acid	NLC	Cetyl palmitate soy lecithin S100, 0.1 mL of octyldodecanol, and 400 mg of VPA,	Emulsion–Solvent Diffusion and Evaporation Method	IN and IP	Wistar rats	NLCs administered via IN showed higher brain:plasma concentration in comparison with NLCs via IP NLCs administered via IN and IP showed similar protective effects to seizure induction	[278]
Ischaemic Stroke	Baicalein	NLC	Tripalmitin, Gelucire^®^ and Hydrogenated soybean phosphatidylcholine 80%	Sonication	IV	Wistar rats	Significantly higher accumulation of the drug from NLCs in all parts of the brain	[280]
	Baicalein	SLN	N/A	N/A	IV	Sprague Dawley rats	A 5.69-fold higher AUC, 6.84-fold higher C_max_ than that of the Baicalein solution. Improved bioavailability of baicalin in cerebral spinal fluid	[281]
	Fas ligand Antibody	PEG-lipid nanoparticles	Monostearin, medium chain triglyceride, Polyethylene glycol monostearate and ODARITC	Solvent Diffusion Method	IV	C57BL/6J wild-type mice	Significant improvements in brain injury and in neurological deficit after ischaemia at significantly lower dose in comparison with regular dl-NB	[282]
	Vinpocetine	Cyclodextrin- NLC	Compritol^®^ 888 ATO and Miglyol^®^ 812N	Water In Oil Emulsification	PO	New Zealand white male rabbits	The relative bioavailability of VP in cyclodextrin-loaded NLC was 592% compared with VP suspension and 92% higher than VP–NLC.	[285]
	Edaravone	Micelle	methoxypoly (ethylene glycol)-b-poly (D,L-lactic acid) (PEG-PLA)	Solvent Evaporation Strategy	IV	ICR mice	The agonistic micelle (EDV-AM) delivered more EDV into brain Ischemia, more rapidly salvaged ischemic tissue EDV-AM showed highest efficiency of in accelerating axonal remodeling and improved functional behaviors	[290]
	mRNA—brain derived neurotrophic factor (BDNF)	Nano -Micelle	N/A	Synthesis Of Block-Copolymers	intraventricular injection	Wistar rats	Prevention of ischemic neuronal death	[292]
	Allium cepa fraction (Ethyl Acetate Fraction)	liposomes	phosphatidylcholine and cholesterol	N/A	IN	Swiss Albino mice	Significant neuroprotection observed at 1/10th the oral dose	[293]
CNS Neoplastic Disease	Temozolomide	SLN	lecithin and Poloxamer 188	Sonication	IV	Rabbits –Pharmacokinetics Kunming mice-Tissue distribution	Higher AUC/dose and the mean residence times in brain and reticuloendothelial cells-containing organs	[297]
	Paclitaxel	LNC	Labrafac^®^ WL 1349, Lipoiïd S75-3 and Solutol^®^ HS 15	Emulsion Inversion Phase Process	N/A	Syngeneic Fischer F344 male rats	Inhibitory effects on efflux pump activity and reduced tumor expansion	[299]
	Camptothecin	SLN	cetyl palmitate, Dynasan^®^ 114 and Witepsol^®^ E85	Oil-In-Water Nanoemulsion Method	IV	Wistar rats	A 6-fold increase in the drug in the brain over the free drug	[300]
	Camptothecin	SLN	cetyl palmitate	High Shear Homogenization and Ultra-Sonication	IV	Wistar rats	A 4.3-fold increase in brain concentration of the drug and detected until 24 h in brain	[301]
	Doxorubicin	SLN	stearic acid, Epikuron 200 and taurocholate sodium salt	Oil-In-Water (O/W) Micro Emulsion	IV	Wistar rats	Significant amount of doxorubicin was detectable in the brain and CSF	[303,304]
	Ferrocenyl diphenol tamoxifen derivative	LNC	Solutol^®^ HS15, Lipoid^®^ and Labrafac^®^	Multi-inversion Phase Processes	SC	Syngeneic Fischer F344 female rats	Significant reduction in both tumor mass and glioma volume	[305]
	Amphetamine	LNC	DOPC, cholesterol and amphetaminylated lipid	Conventional Thin Film Hydration Method	IV	female C57BL/6 mice	LNCs with hexadecyl saturated hydrocarbon chain (16-BACL) showed highest accumulation into the brain, co-solubilized paclitaxel and PD-L1siRNA significantly enhanced the overall survivability of the mice	[306]
	Temozolomide (TMZ)	LNC	Gelucire^®^ 44/14 (solid lipid) and Vit. E (liquid lipid)	High Pressure Homogenization (HPH) Technique	IN	Wistar rats	Higher bioavailability and increased residence time of drug in brain	[308]
	Temozolomide	LNC	Compritol^®^ 888 ATO, Cremophor ELP and soybean phosphatidylcholine	Solvent Diffusion Method	IV	BALB/c nude mice	Significant inhibition (3 times) of tumor growth	[309,310]
	Vincristine (VT) and temozolomide (T)	SLN & LNC	SLN—stearic acid (1 g) and injectable soya lecithin LNC—Compritol^®^ 888 ATO, Cremophor^®^ ELP and soybean phosphatidylcholine	Solvent Displacement Technique	IV	BALB/c nude mice	VT-NLCs inhibited tumor growth over 80%, followed by VT-SLNs (56%) and T-NLCs (70%)	[314]
	Temozolomide (gene-loaded)	(TMZ/DNA-LNCs)	Compritol^®^ 888 ATO, Cremophor^®^ ELP and soybean phosphatidylcholine	Solvent Diffusion Method	IV	BALB/c nude mice	TMZ/DNA-NLCs inhibited tumor growth 3.3 times higher than that of free TMZ	[315]

## Data Availability

Not applicable.

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
