# Peer review of "Lipid-Based Nanocarriers for Neurological Disorders: A Review of the State-of-the-Art and Therapeutic Success to Date"

_pharmaceutics, 2022, doi:10.3390/pharmaceutics14040836_

Round 1
Reviewer 1 Report
The review is interesting and comprehensive. However since the main topic is represented by nanocarriers this part is not well showed.
- Since we have several types of nanocarriers it will be more useful to present these entities in terms of their structures (in a figure) and their properties, advantages or disadvantages etc. (in a table). Otherwise is difficult to follow and to assess the data. For example: “Niosomes are microscopic lamellar structures that are similar to liposomes but are produced using non-ionic surfactants [114].” If we see some structures this phrase will have more meaning. Figure 4 is not enough.
- CSF, API abbreviation must be explained before use in text.
- “2.2. Nose-to-Brain Drug Delivery” “The API is subsequently directly transported to the brain along olfactory and trigeminal nerves”: only one reference and the mechanism is not explained. Which percent of APi in getting into the brain and where specifically.
- “Generally, molecules are transported from the nasal cavity to the brain via three path- 168
- ways; intracellular, extracellular and transcellular as shown in Figure 1.1 [45].” : which route are predominant.
- Also figures of the action mechanisms of some medicines used for diseases described in the paper. After all is a review.
- In 2020 was published a review: Progress and perspectives of brain-targeting lipid-based nanosystems via the nasal route in Alzheimer’s disease (https://doi.org/10.1016/j.ejpb.2019.12.014). Therefore the authors must come with supplementary informations
In my oppinion, in absence of these modifications the paper cannot be published
Author Response
The authors would like to thank the reviewer for taking the time to review this article.
Specific Comments
- Since we have several types of nanocarriers it will be more useful to present these entities in terms of their structures (in a figure) and their properties, advantages or disadvantages etc. (in a table). Otherwise is difficult to follow and to assess the data. For example: “Niosomes are microscopic lamellar structures that are similar to liposomes but are produced using non-ionic surfactants [114].” If we see some structures this phrase will have more meaning. Figure 4 is not enough.
We have added a new image to differentiate between the bilayered nanostructures as requested
2. CSF, API abbreviation must be explained before use in text.
We have amended this throughout the document.
3. “2.2. Nose-to-Brain Drug Delivery” “The API is subsequently directly transported to the brain along olfactory and trigeminal nerves”: only one reference and the mechanism is not explained. Which percent of APi in getting into the brain and where specifically.
We have made some additional statements to this section. However, it is important to note that the specific percentages of API getting to the brain are dependent on physicochemical properties of the API and may vary between different API
4.“Generally, molecules are transported from the nasal cavity to the brain via three path- 168
5. ways; intracellular, extracellular and transcellular as shown in Figure 1.1 [45].” : which route are predominant.
We have added sentences to further explain which of the routes predominates.
6. Also figures of the action mechanisms of some medicines used for diseases described in the paper. After all is a review.
We have added a supplementary section to cover all the aspects of the the action mechanisms.
7. In 2020 was published a review: Progress and perspectives of brain-targeting lipid-based nanosystems via the nasal route in Alzheimer’s disease (https://doi.org/10.1016/j.ejpb.2019.12.014). Therefore the authors must come with supplementary information
We have attempted to cover other CNS disorders in addition to Alzheimers. We have, in the introduction, made a statement to highlight specifically the differences this review has to others.
Best,
Bwalya A. Witika, (Ph.D)
Reviewer 2 Report
The authors presented the paper "Lipid-Based Nanocarriers for Neurological Disorders: A Review of the State-of-the-Art and Therapeutic Success to Date"
1) Some more fresh 2-3 years references have to be added to the sections 1 and 2 to present the level of investigation in the area. Moreover, I have found some new 1-2 year papers in the PubMed and google.schoolar (2022, 2021) which material have to added to review. For example, Lipid nanostructures for targeting brain cancer Heliyon 2021
2) It will be good to summarize the material for in vivo experiment in Section 4 in the Table with the references, disorder, drug, etc.
3) Conclusion section. It will be very good to present the possible "recipe" to cure some disorders in the relation to the lipid-based nanoparticles type. Moreover, some speculation of perspectives of each type of lipid-based nanoparticles are required.
Author Response
We would like to thank the reviewer for taking time to give feedback and allow us to make this a better article.
Specific Comments
1) Some more fresh 2-3 years references have to be added to the sections 1 and 2 to present the level of investigation in the area. Moreover, I have found some new 1-2 year papers in the PubMed and google.schoolar (2022, 2021) which material have to added to review. For example, Lipid nanostructures for targeting brain cancer Heliyon 2021
- We have made necessary changes to the document as requested. The new changes appear in track changes to add more value to the article
2) It will be good to summarize the material for in vivo experiment in Section 4 in the Table with the references, disorder, drug, etc.
- We have added a table to the section as requested to summarize the material in Section 4
3) Conclusion section. It will be very good to present the possible "recipe" to cure some disorders in the relation to the lipid-based nanoparticles type. Moreover, some speculation of perspectives of each type of lipid-based nanoparticles are required.
- The authors have added some information as to what possible remedies can be used to circumvent the shortcomings and/or improve the current lipid nanocarriers
Round 2
Reviewer 1 Report
The paper is improved and is worth to be published in this journal
Reviewer 2 Report
Thank you for the revised paper.